# Metal oxide charge transfer complex for effective energy band tailoring in multilayer optoelectronics

Moohyun Kim[1,4], Byoung-Hwa Kwon [2,4], Chul Woong Joo[2], Myeong Seon Cho[1], Hanhwi Jang[1], Ye ji Kim[1], Hyunjin Cho[1], Duk Young Jeon[1], Eugene N. Cho[3✉] & Yeon Sik Jung [1✉]

Metal oxides are intensively used for multilayered optoelectronic devices such as organic light-emitting diodes (OLEDs). Many approaches have been explored to improve device performance by engineering electrical properties. However, conventional methods cannot enable both energy level manipulation and conductivity enhancement for achieving optimum energy band configurations. Here, we introduce a metal oxide charge transfer complex ($NiO:MoO_3$-complex), which is composed of few-nm-size $MoO_3$ domains embedded in NiO matrices, as a highly tunable carrier injection material. Charge transfer at the finely dispersed interfaces of NiO and $MoO_3$ throughout the entire film enables effective energy level modulation over a wide work function range of 4.47 – 6.34 eV along with enhanced electrical conductivity. The high performance of $NiO:MoO_3$-complex is confirmed by achieving 189% improved current efficiency compared to that of $MoO_3$-based green OLEDs and also an external quantum efficiency of 17% when applied to blue OLEDs, which is superior to 1,4,5,8,9,11-hexaazatriphenylene-hexacarbonitrile-based conventional devices.

---

[1] Department of Materials Science and Engineering, Korea Advanced Institute of Science and Technology (KAIST), 291 Daehak-ro, Yuseong-gu, Deajeon 34141, Republic of Korea. [2] Reality Device Research Division, Electronics and Telecommunications Research Institute (ETRI), 218, Gajeong-ro, Yuseong-gu, Daejeon 34129, Republic of Korea. [3] KAIST Institute for NanoCentury, Korea Advanced Institute of Science and Technology (KAIST), 291 Daehak-ro, Yuseong-gu, Daejeon 34141, Republic of Korea. [4] These authors contributed equally: Moohyun Kim, Byoung-Hwa Kwon. ✉email: nmecho@kaist.ac.kr; ysjung@kaist.ac.kr

Metal oxide thin films are considered an attractive material for various functional applications because of their favorable energy band structure, excellent processability, and high stability[1–3]. In particular, this material system is intensively applied to multi-layered optoelectronic devices such as quantum-dot light-emitting diodes (QLEDs) and organic light-emitting diodes (OLEDs) for ensuring efficient charge transport and charge injection into the emission layer[4–10]. Conventional device design approaches usually adopt pristine metal oxides having compatible electrical properties with the target active layer[11–14]. However, with increasing variation of the active layer and demand for higher device performance, pristine metal oxide systems fall short of meeting the electrical property requirements to achieve both excellent charge transport and injection properties[15–17].

The charge transport and injection properties are typically determined by two factors – the charge carrier density of each layer and the energy level configuration in a device. Although these two factors can be enhanced through engineering of material compositions or configurations, conventional techniques usually have fundamental limitations for tailoring both the charge transport and charge injection properties throughout the entire metal oxide layer. For example, the doping method, the representative conventional approach for improving the charge transport, provides insufficient modulation of the energy band structures of light-emitting devices[10,18–20], and has a narrow tuning range with a doping limit of under 20 at. % due to the formation of second phases and unwanted defects in the lattice[19–21]. On the other hand, surface charge transfer doping (SCTD), a recently emerged materials solution, can significantly modulate the energy level of metal oxides by forming acceptors or donors on the surface of the matrix material[22–26]. However, this enhancement effect is limited to a depth of 10 nm or less at the interface, and proved to be effective only in thin film transistors where the charge flows along the channel, while being unsuitable for multilayered optoelectronic applications where charge flows vertically to the region of the modified electronic structure[22–25].

Herein, we propose a heterostructure based metal oxide charge transfer complex composed of two complementary metal oxides for modulating both the energy level and electrical conductivity by inducing effective charge transfer between the two metal oxides. This approach is aimed at widening the charge transfer range from the localized interface to the whole film by forming nanodomain dispersed fine heterostructure that constitutes a distinct energy band structure at the nanoscale. As an experimental verification, we selected nickel oxide (NiO) as a p-type matrix and embedded molybdenum trioxide ($MoO_3$) nanoparticles (NPs) as strong p-type dopants with a high work function[27,28]. The $NiO:MoO_3$ NPs heterostructure charge transfer complex ($NiO:MoO_3$-complex) exhibits extensive controllability of the work function in a range of 4.47–6.34 eV and electrical conductivity improvement of up to 2.4 times compared to that of pristine NiO, without $MoO_3$ concentration limit. The $NiO:MoO_3$-complex realizes 43% and 189% increased current efficiency in the green phosphorescent OLED system relative to those of pristine NiO and $MoO_3$ by achieving a well-configured energy band structure for excellent electron-hole charge balance. Furthermore, $NiO:MoO_3$-complex is applied to an additional optoelectronic configuration of blue phosphorescent OLEDs, and their high capability and generality is verified on the basis of excellent performance (32.6 cd $A^{-1}$ and 17% external quantum efficiency (EQE)), which is even higher than that of device using 1,4,5,8,9,11-hexaazatriphenylene-hexacarbonitrile (HATCN).

## Results

### Comparison and design of metal oxide charge transfer complex. Figure 1 compares carrier-density modulation methods

(for metal oxides) – doping, surface charge transfer doping (SCTD), and formation of a heterostructure based metal oxide charge transfer complex (new approach in this work). The conventional doping method (Fig. 1a) enhances charge transport throughout the entire metal oxide film by incorporating dopants into the metal oxide lattice. Although the doping effect increases the charge transport ability throughout the entire film due to vacancy formation and charge compensation, the presence of the dopant does not provide sufficient energy level modulation to tune the charge injection property of the metal oxide. On the other hand, SCTD is able to modify the energy level, and as a result the charge injection property of the metal oxide film, by doping the surface of the matrix material prepared with various thin film deposition methods (Fig. 1b). Through interaction between the matrix material and the dopant layer, the non-identical band position of the two materials promotes charge transfer, which, in turn, modifies the energy level at the interface of the two materials of the SCTD system. However, this phenomenon is localized to a 10 nm thick area at the interface of the bilayer, and therefore is unsuitable for multilayered optoelectronic applications where the charge flows vertically to the charge transfer region of the SCTD system.

The two enhancement methods mentioned above provide either full film property enhancement or energy level modulation, but not both. However, modulating the energy level and charge transport while providing the effect throughout the entire film is important for uniform and efficient charge injection and transport across the multilayer structure. Therefore, we designed a heterostructure charge transfer complex to realize charge transfer doping throughout the entire film by fabricating a matrix with embedded nanoparticles (NPs) (Fig. 1c). The heterostructure consisting of the matrix and the homogeneously distributed NPs forms nanodomains of distinct energy structures at the nanoscale that induce charge transfer at the interface between the two materials. In this regard, charge transfer interfaces can be generated throughout the entire layer, allowing the opportunity to modulate the energy level of the entire film, which cannot be achieved by conventional doping and SCTD. Furthermore, the energy level variation can be controlled by adjusting the NP ratio in the matrix to vary the amount of charge transfer interface, resulting in exact matching of the energy structure required to optimize the performance of the optoelectronic device.

**Fabrication of $NiO:MoO_3$ NPs charge transfer complex**. We propose a charge transfer complex consisting of p-type NiO and n-type $MoO_3$ NPs as the donor and acceptor, respectively, to facilitate charge transfer for controlling the energy levels of the composite film. To verify the charge transfer effect between NiO and $MoO_3$ through depth profile X-ray photoelectron spectroscopy (XPS) and ultraviolet photoelectron spectroscopy (UPS), we designed a NiO (20 nm)/$MoO_3$ (20 nm) bilayer of a SCTD system (Supplementary Fig. 1). As seen from the energy diagram in Supplementary Fig. 1a, the work functions of NiO and $MoO_3$ were estimated to be ~4.5 eV and ~5.5 eV, respectively, which correspond to a sufficient energy level difference to promote charge transfer at the interface[4,29]. The shift in the work function around the $NiO/MoO_3$ interface with a width of about 10 nm (Supplementary Fig. 1c, d) provides evidence of charge transfer between NiO and $MoO_3$.

Therefore, in order for NiO and $MoO_3$ to function as a charge transfer complex, the $NiO:MoO_3$ NP heterostructure of $NiO:MoO_3$-complex should be constructed with separate NiO and $MoO_3$ domains in the nanoscale without the formation of alloys, solid solutions, or voids. However, because the Ni-Mo-O system thermodynamically prefers the solid solution or alloy phases[30],

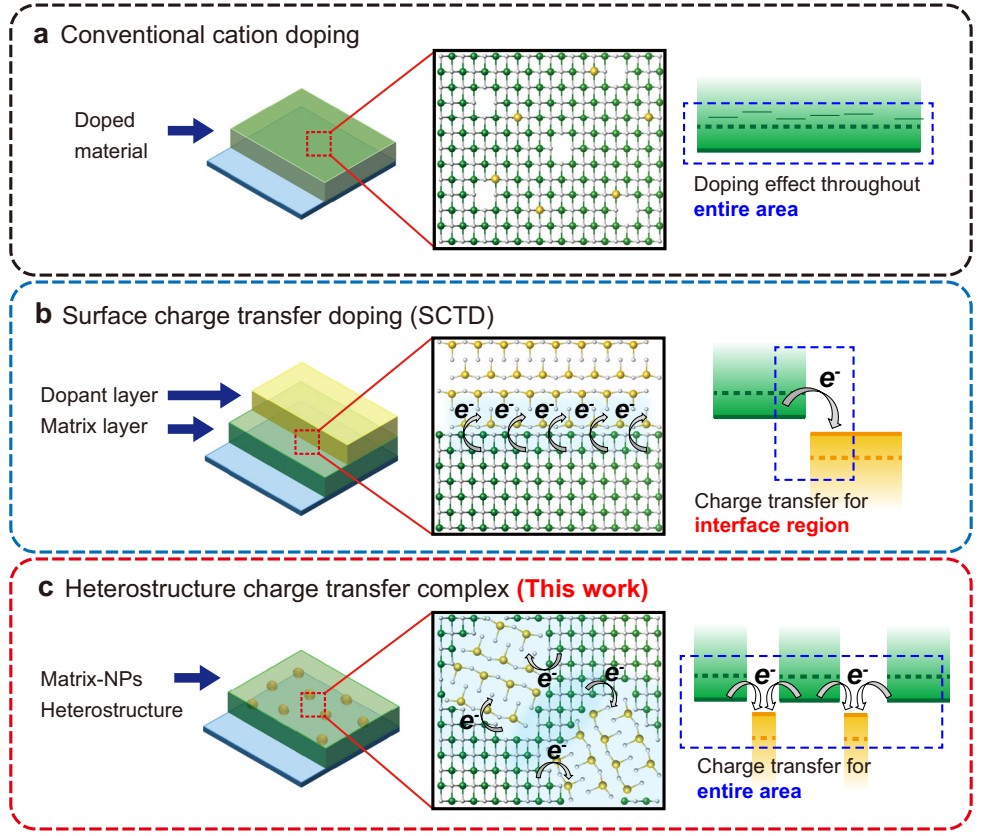

**Fig. 1 Schematics of metal oxide material systems and energy level diagrams that compare conventional doping methods and our heterostructure charge transfer complex formation approach. a** Conventional cation doping, **b** surface charge transfer doping (SCTD), and **c** heterostructure charge transfer complex formation (this work).

the formation of the proposed heterostructure is difficult with typical fabrication methods. Consequently, the heterostructure of the NiO and $MoO_3$ NPs is thermodynamically metastable, and therefore careful design and understanding of the formation of the individual NiO and $MoO_3$ domains is imperative. To form the heterostructure with distinct NiO and $MoO_3$ domain, the $NiO:MoO_3$-complex was fabricated through a nanoparticle complexed sol–gel process. The sol–gel process was chosen because of the extensive controllability it provides over the processing stages (pyrolysis, condensation, and crystallization) in the low-temperature regime to form the wanted materials state[31,32]. To function as an embedded $MoO_3$ domain inside the NiO matrix, the $MoO_3$ NPs (~5 nm size) were synthesized beforehand using a microwave-assisted synthesis method (Supplementary Fig. 2)[33,34]. Surface roughness investigated by atomic force microscopy (AFM) of each pristine and complex films exhibit excellent smoothness and uniformity. The root mean square (RMS) roughness values of all the samples were less than 0.5 nm, for example, an RMS roughness value of 0.17 nm for $NiO:MoO_3$ 50 at. % (Supplementary Fig. 3).

Systematic temperature XRD studies were performed on the sol–gel processed NiO, $MoO_3$ NPs, and $NiO:MoO_3$-NP complex to find the temperature window for forming the designed $NiO:MoO_3$-complex. XRD measurements of the sol–gel processed NiO (Fig. 2a) and $MoO_3$ NPs (Fig. 2b) present patterns corresponding to NiO (JCPDS Card No.47-1049) and $MoO_3$ (JCPDS Card No.05-0508), which were annealed at 250 °C and 350 °C, respectively. At 250 °C, the sol–gel processed NiO starts to undergo crystallization, while at 350 °C, the $MoO_3$ NPs undergo sintering and additional crystallization. The $NiO:MoO_3$-NPs complex (Fig. 2c) shows NiO peaks at 300 °C and $NiMoO_4$ alloy

(JCPDS Card No.45-1042) peaks at 400 °C. These characterization results indicate that the sol–gel process remains in the pyrolysis and condensation step for insufficient heat conditions under 250 °C while the NiO phase and $NiMoO_4$ alloy are formed at above 350 °C (Supplementary Fig. 4). Therefore, we adopted a moderate annealing temperature of 300 °C to promote only NiO crystallization in the complex while minimizing change in the $MoO_3$ NPs (Fig. 2d).

High-resolution transmission electron microscopy (HRTEM) was performed to characterize the $NiO:MoO_3$ NPs heterostructures with varied $MoO_3$ NP compositions of 0%, 15 at. %, 30 at. %, and 50 at. %. The TEM images of the pristine NiO sample (Fig. 2e) exhibit 0.239 nm and 0.208 nm spacing distance (d-spacing), which correspond to NiO (111) and NiO (200) planes, respectively[35,36]. The HRTEM images (Fig. 2f–h) demonstrate that $MoO_3$ NPs of ~5 nm size (yellow dashed circle, 0.359 nm d-spacing of $MoO_3$ (001) space)[37,38] are homogeneously incorporated in the NiO matrix with a linearly increasing proportion upon increasing the fraction of $MoO_3$ NPs, confirming the formation of clearly distinct NiO and $MoO_3$ domains. Fast Fourier transform (FFT) patterns of each HRTEM image further suggest the formation of separate NiO and $MoO_3$ phases depending on the $MoO_3$ NPs content. Another notable point from the TEM analysis is that NiO at the interface with $MoO_3$ NPs is partially amorphized, which is supported by the broadening of the NiO peaks in the XRD patterns of the $NiO:MoO_3$-complex (Fig. 2h and Supplementary Fig. 5). This phenomenon may contribute to the $Ni^{2+}$ vacancy generation in the NiO lattice, which is directly related to the increase of the hole carrier concentration of NiO. The XRD patterns of $MoO_3$ did not appear in the measurements because of its extremely small size[39].

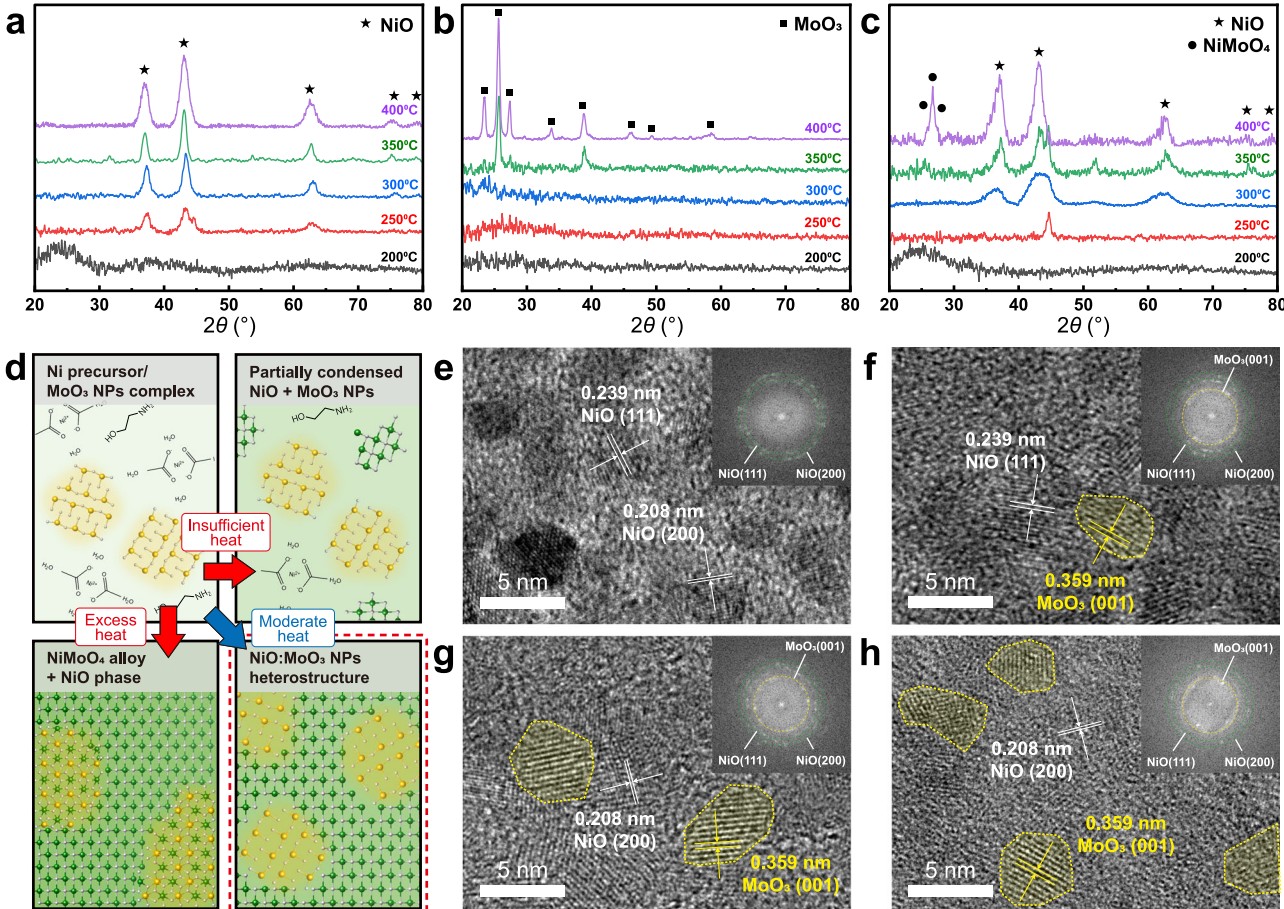

**Fig. 2 Microstructure characterizations of NiO:MoO₃ NPs heterostructure charge transfer complex.** XRD patterns of **a** sol–gel processed NiO, **b** MoO₃ NPs, and **c** MoO₃ 20 at.% complexed NiO with various annealing temperatures. The symbols on the XRD peaks represent NiO (JCPDS Card No.47-1049), MoO₃ (JCPDS Card No.05-0508), and NiMoO₄ (JCPDS Card No.45-1042), respectively. **d** Schematics of the formation of the NiO:MoO₃-complex depending on annealing conditions. High-resolution transmission electron microscopy (HRTEM) images of **e** NiO, **f** NiO:MoO₃ 15 at. %, **g** NiO:MoO₃ NPs 30 at. % and **h** NiO:MoO₃ 50 at. % heterostructures (scale bar: 5 nm). Insets of the HRTEM images correspond to fast Fourier transform (FFT) patterns.

**Charge transfer in the NiO:MoO₃-complex.** To understand the charge transfer behavior at the interfaces between the two phases, an XPS analysis was conducted on the NiO:MoO₃-complex with varying compositions. As shown in Fig. 3a, $Ni^{3+}$ peaks shifted from 855.0 eV to 856.0 eV (dashed line, toward the higher binding energy) with an increasing fraction of MoO₃ NPs in the heterostructure. On the other hand, the Mo $3d_{3/2}$ peak continuously shifted from 235.7 eV to 235.1 eV (to the lower binding energy) for composition change from MoO₃ to NiO:MoO₃ 15 at. % (Fig. 3b, dashed line). These opposing shifts of Ni 2p and Mo 3d peaks as a function of MoO₃ fraction are displayed quantitatively in Fig. 3c.

Peak shifts to the higher binding energies are due to loss of electrons, while peak shifts to lower binding energies are induced by gaining excess electrons. This phenomenon is attributed to the effective electron transfer (charge transfer) between the NiO and MoO₃, as illustrated in Fig. 3d. Depending on the associated ratio of the two phases, the amount of peak shift representing the level of electron loss and gain by charge transfer also varies, demonstrating the level of charge transfer can be manipulated by adjusting the ratio of MoO₃ NP in the complex. To measure the modulation on energy levels from the charge transfer phenomenon, UPS measurements of the NiO:MoO₃-complex were carried out. The work function of the NiO:MoO₃-complex showed a considerable change of 1.87 eV (from 4.47 to 6.34 eV), which is proportional to the MoO₃ NPs fraction in the complex (Fig. 3e).

To compare the range of the work function change in the NiO:MoO₃-complex, a Mo-doped NiO system was also prepared with the same sol–gel fabrication conditions. The UPS measurement data (Supplementary Fig. 6) showed insignificant energy level shifts with increasing Mo concentrations in the complex, presumably due to the lack of charge transfer between the solid solution phase dopants and the NiO matrix, which is also supported by the smaller shift and change in the XPS spectra. This shows that the heterostructure charge transfer complex is a viable approach for modulating the energy level of the entire matrix through charge transfer by adjusting the amount of MoO₃ NPs to the NiO:MoO₃-complex.

Next, the electrical conductivity of the NiO:MoO₃-complex was evaluated by current–voltage measurements of the ITO/NiO:-MoO₃-complex/Au structure (Fig. 3f). Compared to that of pristine NiO, the current density was improved by 79% for 15 at. % MoO₃ NPs and continuously increased up to 140% with increasing MoO₃ NP ratio. It needs to be noted that the log–log plot of the current density–voltage ($J$–$V$) shown in Fig. 3f follows the Ohmic conduction mechanism, which is dominated by electrical conductivity. The Hall effect measurement also provides a consistent trend of increasing conductivity for higher Mo fraction resulting from the enhanced carrier concentration (Supplementary Fig. 7). The increase in the conductivity is presumed to derive from further creation of charge carriers as a consequence of the charge transfer and $Ni^{2+}$ vacancies formation

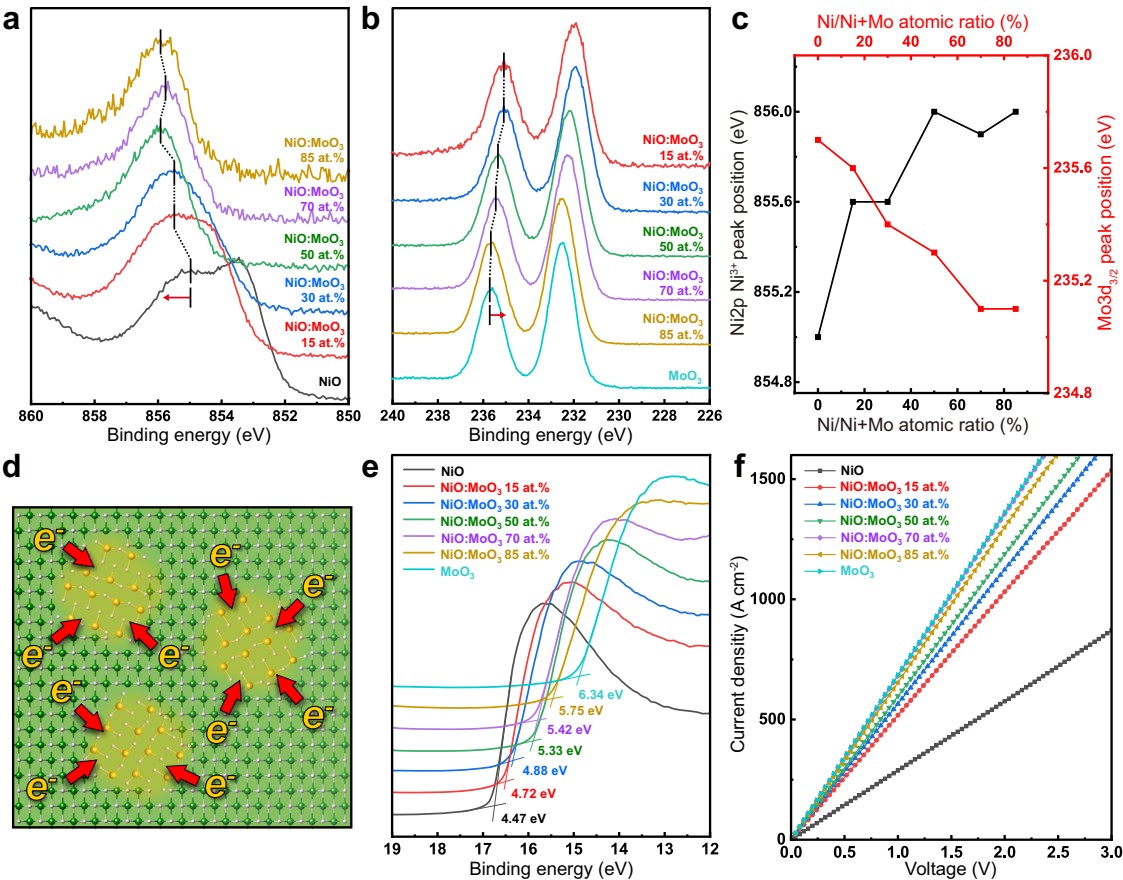

**Fig. 3 X-ray photoelectron spectroscopy studies and electrical property characterization of NiO:MoO$_3$-complex.** XPS of **a** Ni 2*p* and **b** Mo 3*d* narrow scan spectra of NiO:MoO$_3$-complex. Dashed lines refer to peak position shifts by the MoO$_3$ NP fraction. **c** Peak position shift of Ni 2*p* Ni$^{3+}$ and Mo 3*d*$_{3/2}$ with varying compositions of the NiO:MoO$_3$-complex. **d** Schematic of electron charge transfer between the MoO$_3$ NP and NiO interface. **e** Secondary cut-off spectra of ultraviolet photoelectron spectroscopy (UPS) and **f** *I–V* curves for NiO:MoO$_3$-complex.

during the fabrication process. Vacancy generation is supported by the deconvoluted Ni 2*p* XPS spectra of the NiO:MoO$_3$-complex, which can be mainly divided into the Ni$^{2+}$ peak of stoichiometric NiO and the Ni$^{3+}$ peak of off-stoichiometric NiO containing Ni$^{2+}$ vacancies, respectively (Supplementary Fig. 10). The Ni$^{3+}$ peak ratio increases with an increasing fraction of MoO$_3$ NPs in the complex, which is relevant to increased carrier generation in the complex resulting in increased current density and the charge transport ability. The result of vacancy creation according to the MoO$_3$ NPs ratio is also consistent with the structural analysis results from the HRTEM and XRD measurements shown in Fig. 2.

**Performance improvement of NiO:MoO$_3$-complex based OLEDs.** We applied the NiO:MoO$_3$-complex as an hole injection layer (HIL) in green phosphorescent OLED device to verify the ability of energy level modulation to improve device performance (Fig. 4). The current density–voltage–luminance (*J–V–L*) characteristics were measured for various HILs such as pure NiO, 10 at.% to 30 at.% of NiO:MoO$_3$ complex, and pure MoO$_3$ with devices having the energy configuration shown in Fig. 4a. Overall, the current density and luminance decreased with an increasing MoO$_3$ NP fraction in the NiO:MoO$_3$-complex HIL (Fig. 4b). However, the pure MoO$_3$ HIL device showed slightly increased current density compared to the NiO:MoO$_3$ 30 at.% HIL, due to charge imbalance from the inadequate electron-blocking ability of MoO$_3$. Compared to the current efficiencies of NiO (11.4 cd A$^{-1}$) and MoO$_3$ devices (5.6 cd A$^{-1}$) at 1000 cd m$^{-2}$, the NiO:MoO$_3$-

complex showed improved current efficiency, achieving 16.3 cd A$^{-1}$ (MoO$_3$ 30 at.%, at 1000 cd m$^{-2}$), corresponding to 43% and 189% improved current efficiency compared to the NiO and MoO$_3$ HILs. Each device showed consistent green electroluminescence spectra with peaks at 514 nm and CIE 1931 color coordinates of (0.28, 0.64), as shown in Fig. 4d and Supplementary Fig. 11.

The notably improved OLED performance is attributed to the energy level modulation and optimization of the HILs. The NiO:MoO$_3$ complex achieved an excellent electron-hole charge balance by the well matched energy level configuration, which was difficult to achieve with the pure NiO and MoO$_3$ HILs due to their inappropriate energy structure alignment. For a better understanding, Fig. 4e–g show the energy level diagrams of the OLED multilayer structure with NiO, NiO:MoO$_3$-complex, and MoO$_3$. For the NiO or MoO$_3$ HIL, the injection energy level is located at levels that are too shallow or deep and thus result in unfavorable energy structure configurations, as shown in Fig. 4e and g. Therefore, either holes or electrons dominate the structures, resulting in a poor charge balance that leads to low device efficiency. The NiO:MoO$_3$-complex, on the other hand, is able to modulate the energy level to form a well-defined energy structure configuration and effective electron-hole balance, which enhances the overall performance of the device (Fig. 4f).

**High capability and generality of NiO:MoO$_3$-complex.** Finally, the NiO:MoO$_3$-complex was applied to another optoelectronic configuration to investigate its generality as a HIL. Blue

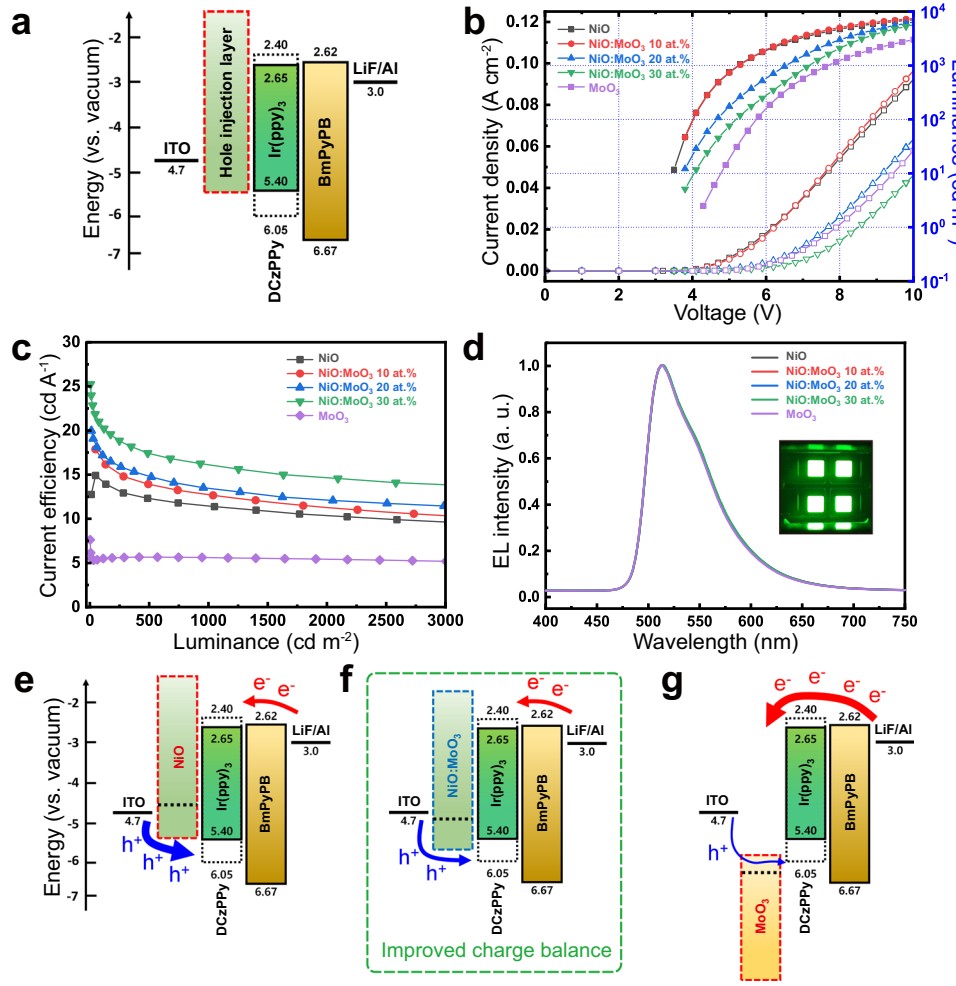

**Fig. 4 Characterization of green phosphorescent OLEDs. a** Energy level diagram of green phosphorescent OLED device. **b** Current density–voltage–luminance curves, **c** luminance-current efficiency curves, and **d** normalized electroluminescence for NiO, NiO:MoO$_3$-complex, and MoO$_3$ based OLEDs. The inset image in **d** shows an operating OLED device. Energy level and charge balance diagrams of multilayer structure configurations with **e** NiO HIL, **f** NiO:MoO$_3$-complex HIL, and **g** MoO$_3$ HIL.

phosphorescent OLEDs were demonstrated, with a structure of ITO/HIL/TAPC/TCTA-DCzppy:Firpic/BmPyPB/LiF/Al (Fig. 5a). A TAPC HTL with high triplet energy and a TCTA-DCzppy:-Firpic double EML were introduced with the purpose of inducing more efficient exciton formation and high device performance. We also evaluated HATCN besides NiO:MoO$_3$-complex as a HIL, which is widely used in various OLED structures, to clarify the generality of the NiO:MoO$_3$-complex. The NiO:MoO$_3$-complex based devices show a similar J–V–L trend to those of the green phosphorescent OLED results above, revealing decreasing in current density and luminance with an increasing ratio of MoO$_3$ NPs (Fig. 5b). However, the overall device performance and efficiency roll-off characteristics were notably improved when compared to green phosphorescent OLEDs. In terms of device performance, high current efficiency of 32.6 cd A$^{-1}$ at 500 cd m$^{-2}$ (~17% of EQE) was achieved in the NiO:MoO$_3$ 10 at.% device which is even higher than that of the HATCN device (27 cd A$^{-1}$ at 500 cd m$^{-2}$, ~14% of EQE) (Fig. 5c and Supplementary Fig. 12). The detailed plots of each green and blue OLEDs were presented in Supplementary Fig. 13. As shown in Fig. 5d, consistent spectra of blue electroluminescence with 472 nm peak were observed for all devices. There were only slight spectrum shift in a broad range of 470–580 nm. In addition, it is found that the device prepared with the charge transfer complex HIL was advantageous over the other approaches discussed earlier, from

comparisons using a blue phosphorescent OLED system (Supplementary Fig. 14).

Consistent tendencies in J–V–L characteristics across the different optoelectronic configurations indicate a definite operation of the NiO:MoO$_3$-complex with effective control of the electron-hole charge balance. Furthermore, we found that optimal device performances was obtained at a different MoO$_3$ fraction for each device structure. This implies that optimum device structures do not always result from a single specific ratio of NiO:MoO$_3$-complex. Hence, the NiO:MoO$_3$-complex has huge potential for diverse optoelectronics such as a QLEDs (Supplementary Fig. 15) or PeLEDs, based on its general applicability due to the facile energy band tailoring ability.

## Discussion

In summary, we introduced the formation of a heterostructure charge transfer complex as a unique approach to simultaneously modulate energy levels and enhance the electrical conductivity of a metal oxide system. A NiO matrix incorporating MoO$_3$ nano-particles of a few-nm size was demonstrated as an example of a heterostructure charge transfer system. In the NiO:MoO$_3$-complex, the energy level can be controlled by varying the MoO$_3$ NPs fraction in the complex because the total interfacial area defines the extent of charge transfer. XRD and HRTEM analyses revealed clearly distinct NiO and MoO$_3$ phases, suggesting that their

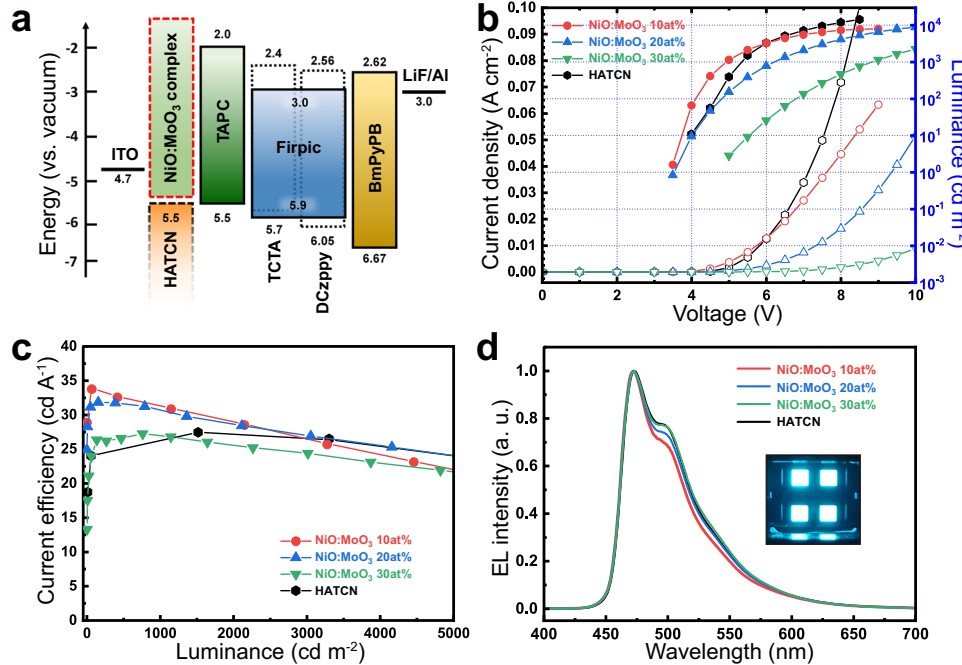

**Fig. 5 Characterization of blue phosphorescent OLEDs. a** Energy level diagram of blue phosphorescent OLED device. HATCN was performed as a comparison HIL with NiO:MoO₃-complex. **b** Current density–voltage–luminance curves, **c** luminance-current efficiency curves, **d** normalized electroluminescence for NiO:MoO₃-complex and HATCN based OLEDs. The inset image in **d** shows an operating OLED device.

individual energy structures are maintained at the nanoscale. Consequently, charge transfer between NiO and MoO₃ was effectively induced throughout the whole heterostructures, extensively modulating their work function from 4.47 to 6.34 eV. The electrical conductivity was also enhanced by 2.4 times relative to that of pristine NiO through effective charge transfer and Ni$^{2+}$ vacancy generation at the NiO-MoO₃ NP interfaces. This energy band modulation ability leads to substantially improved current efficiency in green phosphorescent OLEDs with NiO:MoO₃ 30 at.% HIL, presenting 43% and 189% higher current efficiency than that of pristine NiO and MoO₃ HILs due to well-defined energy band structure formation. Furthermore, blue phosphorescent OLEDs were also demonstrated to verify the high capability and generality of the NiO:MoO₃-complex, showing 32.6 cd A$^{-1}$ current efficiency and 17% EQE with the NiO:MoO₃ 10 at.% HIL, outperforming even the HATCN HIL device. Our strategy overcomes the limitations of conventional metal oxide enhancement methods by delocalizing charge transfer from the surface to the entire film, suggesting an alternative direction for applying metal oxide heterostructures to enhance the performance of multi-layered optoelectronic devices to a great extent.

## Methods

**Materials**. Nickel(II) acetate tetrahydrate ((CH₃CO₂)2Ni·4H₂O, 99%) was purchased from ACROS Organics. Bis(acac)dioxomolybdenum(VI) ((CH₃COCH=C(O⁻)CH₃)₂MoO₂), ethanolamine (NH₂CH₂CH₂OH, ≥99.0%), 2-methoxyethanol (CH₃OCH₂CH₂OH, anhydrous, 99.8%), and hydrogen peroxide solution (H₂O₂, 30 wt%) were supplied by Sigma-Aldrich. Ethanol (C₂H₅OH, ≥99.9%) was prepared from Merck Millipore. All the organic chemicals were used as received without any further purification.

**Synthesis of NiO precursor solution and MoO₃ nanoparticles**. To obtain a 0.1 M NiO precursor solution, 0.5 mmol of nickel(II) acetate tetrahydrate, 30 µl of ethanolamine, and 5 ml of ethanol were blended and stored at 60 °C for 12 h with stirring. MoO₃ nanoparticles were obtained by a microwave-assisted synthesis method. First, 0.5 mmol of molybdenum dioxide bis(acetylacetonate) (C₁₀H₁₆MoO₆) in 5 ml of 2-methoxyethanol was placed in a microwave reactor and heated by a microwave of 2.54 GHz frequency for 60 s. The synthesized MoO₃ nanoparticles solution had a yellowish color. MoO₃ nanoparticles have a mono-disperse distribution with a 2.9 nm median (Supplementary Fig. 2). The NiO

precursor and MoO₃ nanoparticles solutions were then purified by using a PVDF (0.45 µm pore size) filter.

**Preparation of NiO:MoO₃-complex films**. First, 0.4 mol of hydrogen peroxide (H₂O₂) was added to the prepared 0.1 M MoO₃ nanoparticle solution. H₂O₂ acts as a stabilizer from the agglomeration of MoO₃ NPs in a blending solution of Ni precursor and MoO₃ NPs (Supplementary Fig. 16, 17 and Supplementary Movie 1). A MoO₃ NP solution with H₂O₂ was then added to the NiO precursor solutions and blended with a Vortex mixer for 1 min. The mixture solutions were purified with a PVDF (0.45 µm pore size) filter and spin-coated on a cleaned ITO substrate at 3000 rpm for 30 s. The sample was then annealed at 300 °C for 1 h in ambient conditions. The film thickness after spin-coating was 91.4–86.9 nm, and the thickness decreased to 22.4–20.3 nm by the annealing process. Ellipsometry characterization combined with the Cauchy model was introduced to measure the film thickness containing solvent and moisture.

**Characterization of NiO:MoO₃-complex films**. Scanning electron microscope (SEM) measurements were carried out to obtain images with a SU8230 (HITA-CHI). Structural characteristics of NiO, NiO:MoO₃ NPs complex, and MoO₃ NPs were investigated by transmission electron microscopy (HRTEM: JEM-2100F by JEOL, and HAADF-STEM: Talos F200X by FEI). Ultraviolet photoelectron spectroscopy (UPS) measurements were conducted with an Axis-Supra using a He I discharge lamp that has 21.2 eV of photon energy under 9 V bias. The XRD patterns were obtained from an X-ray diffractometer (D/MAX-2500, RIGAKU) using Cu Kα X-ray (λ = 1.542 Å). Atomic force microscopy (AFM) was implemented with a XE-70 Park Systems in non-contact mode with a SSS-NCHR cantilever. A Shimadzu UV-1800 spectrophotometer was employed to obtain the transmittance of the NiO:MoO₃-complex films. X-ray photoelectron spectroscopy (XPS) measurements were performed using a K-alpha (Thermo Fisher Scientific) with automated monochromatic X-ray source Al−Kα 1486.7 eV X-ray photons. Current–voltage curves were measured by source-measure unit (Keithley 238). Electrical resistivity (conductivity), Hall mobility, and carrier concentration were measured by Hall measurement with the van der Pauw method (HMS Model 8407 including high resistance configuration, Lake Shore Cryotronics Inc.).

**Fabrication and measurements of OLEDs**. Green phosphorescent OLEDs were fabricated with a multilayer structure as follows: indium-tin-oxide (ITO, 70 nm)/ HIL (20 nm)/DCzPPy:Ir(ppy)₃ (20 nm:7%)/BmPyPB (40 nm)/LiF (1 nm)/Al (100 nm). The patterned ITO glass substrate was cleaned by sonication in ethanol, acetone, and isopropanol for 20 min, respectively. After UV-ozone treatment of the ITO substrate for 15 min, the NiO:MoO₃-complex (HIL) was spin-coated on the substrates at 3000 rpm for 30 s and then baked at 300 °C for 1 h in ambient conditions. Dczppy:Ir(ppy)₃ and the BmPyPB electron transport layer (ETL) were thermally deposited under ultrahigh vacuum (P = 10⁻⁸ mbar) with thickness of 20

and 40 nm, respectively (deposition rate <1.0 nm·s$^{-1}$). For the electron injection layer, LiF (1 nm) was formed by thermal evaporation on the ETL. Finally, for the cathode layer, an aluminum layer of 100 nm thickness was thermally evaporated with a deposition rate of 0.3 nm·s$^{-1}$. Blue phosphorescent OLEDs were fabricated with a multilayer structure of ITO/HIL/TAPC (20 nm)/TCTA:Firpic (10 nm:7%)/ Dczppy:Firpic (10 nm:10%)/ BmPyPB (55 nm)/ LiF(1 nm)/ Al(100 nm). Processes except HIL deposition were sequentially thermally evaporated in the order of TAPC, TCTA:Firpic, Dczppy:Firpic, BmPyPB, LiF, and Al in ultrahigh vacuum ($P = 10^{-8}$ mbar). The current–voltage–luminance, current efficiency, and electro-luminescence spectra were measured using a source-measure unit system (Keithley 238) and a goniometer-equipped spectroradiometer (Minolta CS-2000). EQE was calculated by using the assumption of a Lambertian distribution of the emission.

## Data availability

The data that support the findings of this study are available from the corresponding author upon reasonable request.

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

## Acknowledgements

This research was supported by the Global Frontier Program through the Global Frontier Hybrid Interface Materials (GFHIM) (2013M3A6B1078874), ICT and Creative Materials Discovery Program (NRF-2016M3D1A1900035), and Nano·Material Technology Development Program (NRF-2021M3H4A3A01062963) of the National Research Foundation of Korea (NRF) funded by the Ministry of Science. This work was also supported by the ITECH R&D Program of MOTIE/KEIT [Project no. 20012560, Development of material component equipment for inkjet printing in flexible QD-OLED].

## Author contributions

M.K. and Y.S.J conceived the project. M.K. and M.S.C. conducted most of the fabrication and the analysis experiments. M.K., B.-H.K. and C.W.J. contributed to OLED device fabrication and characterizations. Y.J.K. and H.C. contributed to XPS and UPS analysis. H.J. contributed to TEM analysis. M.K., E.N.C., and Y.S.J. wrote most of the manuscript. All authors participated to the discussion of the results and writing of the manuscript.

## Competing interests

The authors declare no competing interests.
