## [Peer Review File · Nature Communications]

Reviewers' Comments:

Reviewer #1:

Remarks to the Author:

Kim et al. have developed a NiO:MoO₃ NPs complex for simultaneous adjustment of the energy level and conductivity of the metal oxide film. By using this composite film as an HIL, Ir(ppy)₃-based OLEDs with a current efficiency of ~25 cd/A are obtained. Considering that the device performance is significantly low that those of OLEDs with conventional HIL such as HAT-CN, the advantages of NiO:MoO₃ HIL are very limited. The authors should clarify the advantages of NiO:MoO₃ over conventional HILs and justify their statements. It would be better to compare the device performance and the hole injection capability of NiO:MoO₃ with those of conventional HIL. Other major issues are:

(1) The inorganic NiO:MoO₃ HIL could be applied to QLED rather than OLED. Did the author try its application as HIL for QLED?

(2) Fig. 3 (f), the authors claimed that the NiO:MoO₃ NPs complex can enhance electrical conductivity; however, as the content of MoO₃ NPs increases, the work function of the composite film could also increase, which also contributes to the improvement of the current density of the device (ITO/NiO:MoO₃-complex/Au). The authors should discuss the influence of workfunction on the current density and see whether the change of J-V is dominated by the improvement of conductivity or the reduction of injection barrier.

(3) The hole concentration, the hole mobility and the injection barrier of the NiO/MoO₃ should be measured to see which one is the dominant factor for the change of current.

(4) Fig. 4(b), why does the pure-NiO (with lowest conductivity and work function) based device shows the highest current density and brightness?

(5) Fig. 4 (c), the author attributes the efficiency improvement of NiO/MoO₃ 30 at%-based device to the improved charge balance ; however, why its efficiency roll off is more serious than that of MoO₃-based or NiO-based devices?

(6) Because there is no HTL in the device, the device may exhibit large leakage current that may be the reason for the low efficiency. Therefore, I suggest the authors to plot the current density on a log scale in Fig. 4(b) to analyze the leakage current of the devices at low current density.

(7) A good HIL should have low surface roughness, and thus I suggest the authors to investigate the surface morphology especially the surface roughness of the NiO:MoO₃.

(8) Fig. 4 (b), why NiO:MoO₃ with higher work function and higher conductivity shows less hole injection current?

Reviewer #2:

Remarks to the Author:

This paper reported an approach to improve the device performance of multilayer OLEDs by using a charge transfer complex (NiO:MoO₃-complex) as the hole injection layer. Although systematical characterizations on the structures and the charge transfer processes of NiO:MoO₃-complexes have been conducted in this research, the statement for performance improvement of the OLEDs based on NiO:MoO₃-complexes is unconvincing. Also, the authors didn't demonstrate the generality of this approach for improving the performance of OLEDs. Therefore, I cannot recommend the publication of this manuscript in Nat. Commun.

1. In Figure 3f, it is noticed that the pure MoO₃ layer exhibited the highest conductivity among these complexes. How could the authors exclude the possibility that the improvement of the complex conductivity with the increased content of MoO₃ is due to the high conductivity of MoO₃?

2. The authors claimed that "the NiO:MoO₃ 30 at. %-complex HIL showed a substantially improved current efficiency (43% and 189%, respectively), due to well-defined energy structure formation." However, the substantial improvement comes from self-comparison instead of the comparison with results reported in other works. Therefore, it is not so convincing. Especially, the performance of OLEDs with pristine NiO or MoO₃ HIL is very low and no statistic data is presented here.

3. The authors claimed that they could induce effective energy level modulation over a wide work

function range of 4.47-6.34 eV. How about the general applicability of using the NiO:MoO₃-complex to improve the performance of OLEDs based on other luminescent materials besides DCzppy-Ir(ppy)₃?

Reviewer #3:

Remarks to the Author:

From my point of view, this manuscript shows results which are worthy of publication in Nature Communications. However, some corrections had to be made previous to the acceptance of the paper for publication. These changes are related with the next comments:

1- I think that "complex" is not the correct word that should be used to define these materials. They are composite materials formed by nanoparticles immersed in a matrix. Therefore, I think "complex" should be changed by "composite. Actually, the material is not a NiO:MoO₃ NPs "heterostructure" (not different layers in the film), but a composite thin film material of a NiO matrix with MoO₃ NPs.

2- Pg.10. The efficiencies obtained in these materials should be compared with those of materials of these metal oxides prepared by the doping method and with the SCTD. It would be good to show a table or graphic showing results obtained using other preparation concepts compared with the results obtained in this work for different NiO/MoO₃ NPs ratios.

3. Fig.4. Differences in the results of electroluminescence obtained for NiO, NiO_MoO₃-complex and MoO₃ are not clearly observed Fig 4d. Modify this figure to see at a first sight the differences among the materials.

4. Discussion. Actually, this is the conclusion section. Here, discussion of the results is not made. It is made in the previous section of Results. Therefore, you can do two things if the format of the journal allow you to do it: 1st. Include two different sections: "Results" and "Discussion", separating and moving the discussion of the results now done in the "Results" section to the "Discussion" section. 2nd. Changing the caption of the Results section by "Results and Discussion". And then, the "Conclusions".

5. Methods. In "Synthesis of NiO precursor solution and MoO₃ nanoparticles", Results should be provided in this section about the characterization of the MoO₃ NPs. Mainly, average particle size and distribution of particle sizes. Crystallinity is correctly shown in Fig 2 by XRD, as a function of the annealing temperature.

6. Methods. In "Preparation of NiO:MoO₃-complex films", How do you assure that agglomeration of the MoO₃ NPs is not produced? A simple blending of the NiO solution and the MoO₃ NPs is difficult that avoid nanoparticles agglomeration. Indicate this in this section. Also here, Indicate the thickness of the layer that is obtained after the deposition of the mixture solutions.

7. Methods. In "Fabrication and measurements of OLEDs". Spelling mistakes, it should be 300 °C. Also, here indicate the thickness of the layer after the annealing.

■ Point-by-point responses to the reviewers' comments.

We provide point-by-point responses to the reviewers' in-depth comments, and the updated text in the revised manuscript is marked in blue.

Reviewer #1

[General remarks] *Kim et al. have developed a NiO:MoO₃ NPs complex for simultaneous adjustment of the energy level and conductivity of the metal oxide film. By using this composite film as an HIL, Ir(ppy)₃-based OLEDs with a current efficiency of ~25 cd/A are obtained. Considering that the device performance is significantly low that those of OLEDs with conventional HIL such as HAT-CN, the advantages of NiO:MoO₃ HIL are very limited. The authors should clarify the advantages of NiO:MoO₃ over conventional HILs and justify their statements. It would be better to compare the device performance and the hole injection capability of NiO:MoO₃ with those of conventional HIL. Other major issues are:*

[General response] We appreciate the reviewer's insightful comments that enabled us to greatly improve our manuscript. As the reviewer pointed out, we agree that showing the advantages of the NiO:MoO₃ complex system more clearly will help readers understand our manuscript more convincingly. Therefore, additional OLED structures based on a different luminescent material (Firpic, blue) were demonstrated to support the advantages of the NiO:MoO₃ complex in previously described OLED results (Ir(ppy)₃, green). In this regard, HATCN and other metal oxide enhancing approaches were compared to the NiO:MoO₃ complex. Further studies of the NiO:MoO₃ complex thin film were also conducted. We hope the revised version of our manuscript successfully delivers the concept of our work to the reviewer and readers of the journal.

[Comment 1] *The inorganic NiO:MoO₃ HIL could be applied to QLED rather than OLED. Did the author try its application as HIL for QLED?*

[Response 1] We are grateful to the reviewer for providing helpful comments regarding the application of our NiO:MoO₃ HIL for quantum-dot light emitting diodes (QLEDs). Demonstrating optoelectronics based on a different luminescent system such as quantum dots will demonstrate the generality of our charge transfer complex. To this end, we fabricated and characterized green-emitting QLED devices. The device structure was ITO/NiO:MoO₃ complex/PVK/CdSe QDs/ZnO/Al, which is one of the typical normal structure of QLEDs. As shown in Supplementary Fig. 15, a moderate level of *J-V-L* characteristic along with a peak current efficiency of 2.95 cd A⁻¹ at 1000 cd m⁻² was obtained. Also, a narrow green emission of 525 nm supports the fine operation of the NiO:MoO₃ complex as a HIL in QLEDs. These points indicate that the improved *J-V-L* characteristic and performance are attainable by the additional optimization. For example, we expect that better device performance can be achieved by gradient energy band formation via stacking charge transfer complex thin films. In response to the reviewer's comment, Supplementary Information was updated to include the application of the NiO:MoO₃ complex to a QLED device.

[Modifications in the manuscript]

(Supplementary Information)

Supplementary Fig. 15. Characterization of quantum dot light emitting diodes (QLEDs). (a) Energy level diagram of multilayer QLED device. (b) Current density-voltage and luminance curves as a function of applied voltage, (c) luminance-current efficiency curves, (d) normalized electroluminescence for NiO:MoO₃ complex QLEDs. The dashed line in (b) is the polynomial fit curve of luminance. The inset image in (d) shows an operating QLED device.

Methods

QLED device fabrication.

The ITO-patterned glass substrates (resistivity: $< 20 \text{ ohm square}^{-1}$) were cleaned by sonication with acetone, ethanol, and isopropyl alcohol for 30 min each, followed by UV-O₃ treatment for 20 min. The QLED devices were fabricated by spin-coating NiO:MoO₃ complex, PVK (TCI), QDs (Uniam), and ZnO in order, except for the Al cathode. After spin-casting of NiO:MoO₃ 10 at.% complex (3000 rpm, 30 s), the sample was annealed at 300 °C for 1 hour under ambient conditions and was then moved to a N₂ gas filled glove box to form the next layers. PVK (8 mg mL⁻¹ in chlorobenzene) was spin-coated at 3000 rpm for 60 s and baked at 180 °C for 30 min, followed by QD solution (15 mg mL⁻¹ in octane) coating at 3000 rpm for 30 seconds and baking for 30 min at 80 °C. ZnO NPs (30 mg mL⁻¹ in 2-methoxyethanol)

were deposited by spin-coating (3000 rpm for 60 seconds) onto the QD layer. Finally, a 120 nm Al cathode was thermally evaporated under high vacuum conditions (10^{-8} mbar).

Synthesis of ZnO NPs.

To synthesize ZnO NPs, 1.8 mmol of zinc acetate dihydrate ($\text{Zn(OAc)}_2 \cdot 2\text{H}_2\text{O}$) was loaded in a 100 mL flask with 20 ml of ethanol. The Zn precursor solution was stirred for 1 h at 80 °C. After adding 2.8 mmol of lithium hydroxide monohydrate ($\text{LiOH} \cdot \text{H}_2\text{O}$), the flask was sonicated for 1 h to synthesize ZnO NPs. Finally, the synthesized ZnO NP solution was purified by centrifugation with hexane mixed into the solution.

[Comment 2] *Fig. 3 (f), the authors claimed that the NiO:MoO₃ NPs complex can enhance electrical conductivity; however, as the content of MoO₃ NPs increases, the work function of the composite film could also increase, which also contributes to the improvement of the current density of the device (ITO/NiO:MoO₃-complex/Au). The authors should discuss the influence of work function on the current density and see whether the change of J-V is dominated by the improvement of conductivity or the reduction of injection barrier.*

[Response 2] We appreciate the reviewer's important comment regarding the *J-V* characteristic of the NiO:MoO₃-complex. It is very crucial to clarify the major factor of current density increment in the ITO/NiO:MoO₃-complex/Au device, and thus we measured the *J-V* characteristics with the electrode/metal oxide/electrode structure. Metal oxides such as NiO and MoO₃ between the anode and cathode are known to follow the bulk-limited electronic conduction mechanism due to their very low conductivity and wide bandgap.^{1,2} The *J-V* characteristic of the electrode/metal oxide/electrode system can be divided into four different regimes, which are respectively designated as ohmic, trap-limited space charge limited current (SCLC), trap-filling, and SCLC regimes. In the ohmic regime occurring at a low voltage range, current density obeys Ohm's law ($J = q\mu n_0 \xi = \sigma \xi$) because the intrinsic carrier density of a metal oxide is larger than the injected carrier density from the electrode. In this regard, we could solely determine the effect of conductivity on the current density in the ohmic conduction regime, without considering injection barrier or energy level related factors. Our results for the *J-V* characteristic (Fig. 3f)

were expressed with a log J -log V plot, as shown in Supplementary Fig. 7a, presenting consistent fitted slopes of 1.00, which indicate ohmic conduction of NiO, NiO:MoO₃ complex, and MoO₃. In addition, in response to the reviewer's comment, the electrical resistivity (conductivity) of NiO, MoO₃, and NiO:MoO₃ complex thin films was respectively determined using a conducting Hall effect measurement system. As shown in Supplementary Fig. 7b, NiO was found to have the highest resistivity of 5,390 $\Omega\cdot\text{cm}$ and it drastically decreased to 3,407 $\Omega\cdot\text{cm}$ for the NiO:MoO₃ complex with Mo 15 at.%. As the Mo ratio was further increased, the resistivity gradually reached 2,357 $\Omega\cdot\text{cm}$ (for MoO₃). When comparing the resistivity with the current density at 1 V from Fig. 3f, a fairly similar trend – a steep change appearing at 15 at.% of Mo – was observed, followed by a gradual change depending on the Mo ratio. Therefore, considering the results of Supplementary Fig. 7, we could conclude that the J - V trend in Fig. 3f mainly originated from the enhancement of electrical conductivity of the NiO:MoO₃ complex. In response to the reviewer's comment, we added Hall measurement results of NiO, MoO₃, and NiO:MoO₃ complex films, along with more detailed analyses on the J - V trend.

References

1. Greiner MT, Lu Z-H. Thin-film metal oxides in organic semiconductor devices: their electronic structures, work functions and interfaces. *NPG Asia Materials* **5**, e55-e55 (2013).
2. Chiu FC. A Review on Conduction Mechanisms in Dielectric Films. *Adv Mater Sci Eng* **2014**, (2014).

[Modifications in the manuscript]

(Results, page 10) “It needs to be noted that the log-log plot of J - V shown in Fig. 3f follows the Ohmic conduction mechanism, which is dominated by electrical conductivity. The Hall effect measurement also provides a consistent trend of increasing conductivity for higher Mo fraction resulting from the enhanced carrier concentration (Supplementary Fig. 7). The increase in the conductivity is presumed to derive from further creation of charge carriers as a consequence of the charge transfer and Ni²⁺ vacancies formation during the fabrication process. Vacancy generation is ~”

(Methods, page 16) “Electrical resistivity (conductivity), Hall mobility, and carrier concentration were measured by Hall effect measurement system with the van der Pauw method (HMS Model 8407 including high resistance configuration, Lake Shore Cryotronics Inc.).”

(Supplementary Information)

Supplementary Fig. 7. Electrical conductivity trends of NiO:MoO₃ complex thin film. (a) Log J -log V plot of J - V characteristic (Fig. 3f) in ITO/thin film/Au device of NiO, MoO₃, and NiO:MoO₃ complex. Dashed lines are the linear fit of each device, following the ohmic conduction regime (slope = 1.00). (b) Comparison of resistivity conducted by the Hall measurement system and current density at 1 V from J - V curve (Fig. 3f) of NiO, MoO₃ and NiO:MoO₃ complex thin film. (c) Electrical conductivity, Hall mobility, and carrier concentration of NiO, MoO₃, and NiO:MoO₃ complex thin film obtained from Hall effect measurement.

[Comment 3] *The hole concentration, the hole mobility and the injection barrier of the NiO/MoO₃ should be measured to see which one is the dominant factor for the change of current.*

[Response 3] We agree with the concern raised by the reviewer and the need to verify the dominant factor

underlying the increased current density in Fig. 3f. To further clarify the electrical conductivity enhancement mechanism discussed in Response 2, Hall effect measurement was used. As shown in Supplementary Fig. 7c, the electrical conductivity sharply enhanced from $1.86 \cdot 10^{-4}$ S·cm of NiO to $2.93 \cdot 10^{-4}$ S·cm at NiO:MoO₃ 15 at.%, and increased to $4.24 \cdot 10^{-4}$ for MoO₃. Although the Hall mobility decreased rapidly in the NiO:MoO₃ complex system, the carrier concentration increased to a greater extent, contributing to the increasing trend of electrical conductivity in the NiO:MoO₃ complex thin film. This is noticeable because the results strongly support our model that charge carriers were effectively generated by charge transfer from the NiO and MoO₃ complex structure, in addition to Ni vacancy creation during the fabrication process.

On the other hand, it is also important to check whether the injection barrier (barrier height) has significantly changed for the further confirmation. First, ultraviolet photoelectron spectroscopy (UPS) was conducted to investigate the detailed electronic energy levels (conduction and valence band edge) of the NiO:MoO₃ complex. On the basis of work-function values from secondary cut-off spectra (Fig. 3e), valence band onset spectra, and the bandgap from a Tauc plot (Supplementary Fig. 8), several electronic energy levels were evaluated. In particular, individual valence band onset points for NiO and MoO₃ were estimated separately because the NiO:MoO₃ complex is not an alloy but distinct metal oxides, similar to an organic charge transfer complex.^{1,2} As shown in Supplementary Fig. 8a, valence band onset spectra were classified to the representative valence band orbital of NiO³ and MoO₃^{4,5} to determine valence band shift. Detailed values of each electronic level are described in Supplementary Table 1 and Fig. 7. Peaks at around 0.8 and 2.0 eV were excluded for evaluation of the onset point because they are assigned to MoO₃ gap state, which is unrelated to the original MoO₃ band structure. Also, Supplementary Table 1 summarizes the calculated electronic energy levels and barrier heights at the interface with electrodes from the evaluated values. We found that the electronic energy levels move downwards with increasing MoO₃ fraction in the complex, as a consequence of charge transfer between NiO and MoO₃ (Supplementary Fig. 9). Especially in the system shown in Fig. 3f, greater barrier height between the NiO:MoO₃ complex and electrode was induced since the NiO:MoO₃ complex has deeper energy levels than the work function of the electrodes (4.7 eV of ITO and 5.1 eV of Au). However, the NiO:MoO₃ complex shows improved current density even though it is not beneficial for charge injection. This is because the metal oxide thin film between the electrodes follows Ohmic conduction at the low voltage regime regardless of the energy level formation. We believe that these data support the results in Fig. 3f showing that the *J-V* characteristics clearly present enhanced conductivity of the NiO:MoO₃ complex. Accordingly, in response to the reviewer's comment, supplementary data and corresponding sentences

were added to help the understanding of Fig. 3f.

References

1. Zhong J-Q, *et al.* Energy Level Realignment in Weakly Interacting Donor–Acceptor Binary Molecular Networks. *ACS Nano* **8**, 1699-1707 (2014).
2. Otero R, Vázquez de Parga AL, Gallego JM. Electronic, structural and chemical effects of charge-transfer at organic/inorganic interfaces. *Surface Science Reports* **72**, 105-145 (2017).
3. Hüfner S. Electronic structure of NiO and related 3d-transition-metal compounds. *Advances in Physics* **43**, 183-356 (1994).
4. Vasilopoulou M, *et al.* The Influence of Hydrogenation and Oxygen Vacancies on Molybdenum Oxides Work Function and Gap States for Application in Organic Optoelectronics. *Journal of the American Chemical Society* **134**, 16178-16187 (2012).
5. Wu Q-H, *et al.* Electronic structure of MoO_{3-x}/graphene interface. *Carbon* **65**, 46-52 (2013).

[Modifications in the manuscript]

(Results, page 10) “It needs to be noted that the log-log plot of J - V shown in Fig. 3f follows the Ohmic conduction mechanism, which is dominated by electrical conductivity. The Hall effect measurement also provides a consistent trend of increasing conductivity for higher Mo fraction resulting from the enhanced carrier concentration (Supplementary Fig. 7). The increase in the conductivity is presumed to derive from further creation of charge carriers as a consequence of the charge transfer and Ni²⁺ vacancies formation during the fabrication process. Vacancy generation is ~”

(Methods, page 16) “Electrical resistivity (conductivity), Hall mobility, and carrier concentration were measured by Hall effect measurement system with the van der Pauw method (HMS Model 8407 including high resistance configuration, Lake Shore Cryotronics Inc.)”

(Supplementary Information)

Supplementary Fig. 7. Electrical conductivity trends of NiO:MoO₃ complex thin film. (a) Log J-log V plot of *J-V* characteristic (Fig. 3f) in ITO/thin film/Au device of NiO, MoO₃, and NiO:MoO₃ complex. Dashed lines are the linear fit of each device, following the ohmic conduction regime (slope = 1.00). (b) Comparison of resistivity conducted by the Hall measurement system and current density at 1 V from *J-V* curve (Fig. 3f) of NiO, MoO₃ and NiO:MoO₃ complex thin film. (c) Electrical conductivity, Hall mobility, and carrier concentration of NiO, MoO₃, and NiO:MoO₃ complex thin film obtained from Hall effect measurement.

Supplementary Fig. 8. Ultraviolet photoelectron spectroscopy (UPS) spectra and Tauc plot for energy level evaluation. (a) Valence band onset spectrum of NiO, NiO:MoO₃-complex, and MoO₃. Tauc plot of (b) NiO and (c) MoO₃. Valence band onset points of NiO and MoO₃ were counted separately because the NiO:MoO₃ complex is not an alloy but a distinct metal oxides similar to organic charge transfer complex.

Supplementary Table 1. Electronic energy levels and barrier heights with electrodes of NiO, MoO₃, and NiO:MoO₃ complex.

		NiO	NiO:MoO ₃ 15 at. %	NiO:MoO ₃ 30 at. %	NiO:MoO ₃ 50 at. %	NiO:MoO ₃ 70 at. %	NiO:MoO ₃ 85 at. %	MoO ₃
Work-function (eV)		4.47	4.72	4.88	5.33	5.42	5.75	6.34
NiO	Valence band shift (eV)	0.9	0.7	0.64	0.56	0.6	0.64	-
	Valence band edge (eV)	5.37	5.42	5.52	5.89	6.02	6.39	
	Bandgap (optical, eV)	3.64						
	Conduction band edge (eV)	1.73	1.78	1.88	2.25	2.38	2.75	
MoO ₃	Valence band shift (eV)	-	2.7	2.77	2.92	3	2.96	2.8

	Valence band edge (eV)		7.42	7.65	8.25	8.42	8.71	9.14
	Bandgap (optical, eV)		3.19					
	Conduction band edge (eV)		4.23	4.46	5.06	5.23	5.52	5.95
	NiO E_v – ITO (eV)	0.67	0.72	0.82	1.19	1.32	1.69	–
	ITO – MoO ₃ E_c (eV)	–	0.47	0.24	-0.36	-0.53	-0.82	-1.25
	NiO E_v – Au (eV)	0.27	0.32	0.42	0.79	0.92	1.29	–
	Au – MoO ₃ E_c (eV)	–	0.87	0.64	0.04	-0.13	-0.42	-0.85

Supplementary Fig. 9. Energy diagrams of the NiO, MoO₃, and NiO:MoO₃ complexes with varied MoO₃ composition.

[Comment 4] Fig. 4(b), why does the pure-NiO (with lowest conductivity and work function) based device shows the highest current density and brightness?

[Response 4] In the OLED system, current density is the sum of electron and hole currents, which are governed by the energy barrier and conductivity of the multilayer device structure. In particular, the energy barrier is a major factor of charge carrier injection into the EML, which is determined by the different energy band structures at the interface of the materials rather than the properties of the single material.¹ Consequently, current density along with luminance and corresponding efficiency in our device structure are highly dependent on energy level of applied HIL. NiO has lowest conductivity and work function of 4.47 eV among NiO, MoO₃ and NiO:MoO₃ complexes. However, it provides a relatively

lower barrier height at ITO/HIL and HIL/EML interfaces (Table R1), and therefore NiO HIL device showed the highest value of current density even though NiO had lowest electrical conductivity. Nevertheless, pristine NiO based OLED exhibits a much lower efficiency than those of NiO:MoO₃ complexes due to poor electron-hole charge balance. For NiO:MoO₃ complexes, the work function increases up to 0.41 eV for 30 at.% compared to pristine NiO, and these work function shift causes reduced levels of *J-V-L* characteristic. These tendency of *J-V-L* characteristics were likewise found from additional experiments on different optoelectronic configuration of blue phosphorescent OLEDs (Fig. 5). This is an important point because it is consistent with our results and arguments that tailored electronic properties for particular structure is imperative to obtain excellent device performance, rather than simply high or low value of conductivity or energy level being preferred for a certain structure.

Table R1. Barrier height at interface of anode/HIL and HIL/EML host in green OLED configuration.

	NiO	NiO:MoO ₃ 15 at.%	NiO:MoO ₃ 30 at.%	MoO ₃
Work function (eV)	4.47	4.72	4.88	6.34
NiO E _v – ITO (eV)	0.67	0.72	0.82	-
MoO ₃ E _c – ITO (eV)	-	-0.47	-0.24	1.25
DCzppy – NiO E _v (eV)	0.68	0.63	0.53	
DCzppy – MoO ₃ E _c (eV)	-	1.59	0.99	0.1

Reference

1. Simmons JG. Richardson-Schottky Effect in Solids. *Physical Review Letters* **15**, 967-968 (1965).

[Comment 5] Fig. 4 (c), the author attributes the efficiency improvement of NiO/MoO₃ 30 at%-based device to the improved charge balance ; however, why its efficiency roll off is more serious than that of MoO₃-based or NiO-based devices?

[Response 5] We understand the reviewer’s concern regarding the efficiency roll-off characteristic. The

roll-off can appear to be more significant for the NiO:MoO₃ devices likely due to higher efficiencies; in fact, it has a decreasing trend with increasing Mo content. The efficiencies of the NiO and MoO₃ devices were much lower than that of the NiO:MoO₃ complex, and there is a similar degree of roll-off for all devices. The roll-off characteristics showed an increasing order from NiO to MoO₃ (same order as the work function), exhibiting the worst characteristics in the pristine MoO₃ device. Fig. R1 and Table R2 provide detailed data for the efficiency roll-off characteristics including the critical current density ($J_{90\%}$). $J_{90\%}$ is an index of efficiency roll-off indicating the specific current density where the external quantum efficiency (EQE) drops to 90% of the maximum EQE value.¹

Basically, we designed the device structure without an HTL to display the energy level modulation capability of the NiO:MoO₃ complex more definitively (Fig. 4). As a result, the current efficiency is drastically improved, but the possibility of exciton quenching is promoted due to the direct interface of the inorganic HIL and organic EML. Therefore, we demonstrated an additional device structure to support our claims while resolving the reviewer's concerns. A blue phosphorescent OLED was designed with a structure of ITO/HIL/TAPC/TCTA-DCzppy:Firpic/BmPyPB/LiF/Al. A TAPC HTL with high triplet energy and a TCTA-DCzppy:Firpic double EML were introduced to induce more efficient exciton formation and suppress the exciton quenching, leading to high device performance with improved efficiency roll-off characteristics. Furthermore, characterization of blue phosphorescent OLEDs including a comparison with the conventional HATCN HIL was provided to verify high compatibility and generality of the NiO:MoO₃ complex. As shown in Fig. 5, the device structure, J - V - L characteristic, device efficiency, and spectra of the OLEDs were described in detail. For all devices, efficiency roll-off was significantly improved to the level of the HATCN HIL device, while achieving high current efficiency exceeding 32 cd A⁻¹ (~ 17% of EQE, Supplementary Fig. 13f). Also, consistent tendencies depending on the Mo ratio in the J - V - L characteristics across different optoelectronic configurations clearly indicate improved charge balance by tailoring the energy band. In response to the reviewer's comment, these results are updated in the revised manuscript and Supplementary Information to clarify the efficiency roll-off characteristics.

Fig. R1. Current density-EQE curves of green phosphorescent OLED at lower current density range under 0.002 A m^{-2} (related to Fig. 4).

Table R2. Detailed efficiency roll-off values of green phosphorescent OLED (related to Fig. 4).

HIL	Peak external quantum efficiency (%)	90% of external quantum efficiency (%)	Critical current density ($J_{90\%}$, A cm^{-2})
NiO	4.58	4.12	1.51E-3
NiO:MoO ₃ 10 at.%	5.48	4.93	9.22E-4
NiO:MoO ₃ 20 at.%	6.13	5.52	3.64E-4
NiO:MoO ₃ 30 at.%	7.75	6.98	1.14E-4
MoO ₃	2.34	2.10	6.79E-5

Reference

1. Murawski C, Leo K, Gather MC. Efficiency Roll-Off in Organic Light-Emitting Diodes. *Advanced Materials* **25**, 6801-6827 (2013).

[Modifications in the manuscript]

(Results, page 12) “The detailed plots of each green and blue OLED were presented in Supplementary Fig. 13.”

Fig. 5. Characterization of blue phosphorescent OLEDs. (a) Energy level diagram of blue phosphorescent OLED device. HATCN was performed as a comparison HIL with NiO:MoO₃ complex. (b) Current density-voltage-luminance curves, (c) luminance-current efficiency curves, (d) normalized electroluminescence for NiO:MoO₃ complex and HATCN based OLEDs. The inset image in (d) shows an operating OLED device.

(Methods, page 17) “Blue phosphorescent OLEDs were fabricated with a multilayer structure of ITO/HIL/TAPC(20 nm)/TCTA:Firpic (10 nm:7%)/DCzppy:Firpic (10 nm:10%)/BmPyPB (55 nm)/LiF(1 nm)/Al(100 nm). Processes except HIL deposition were sequentially thermally evaporated in ultrahigh vacuum ($P = 10^{-8}$ mbar), as done for green phosphorescent OLEDs.”

(Methods, page 17) “EQE was calculated by using the assumption of a Lambertian distribution of the emission.”

(Supplementary Information)

Supplementary Fig. 13. Detailed device characteristics of green and green and blue phosphorescent OLEDs. (a) Band structure of green phosphorescent OLED device and (b) logarithmic plot of J - V - L (relative to Fig. 4). (a) Band structure of blue phosphorescent OLED device and (b) logarithmic plot of J - V - L (relative to Fig. 5).

[Comment 6] *Because there is no HTL in the device, the device may exhibit large leakage current that may be the reason for the low efficiency. Therefore, I suggest the authors to plot the current density on a log scale in Fig. 4(b) to analyze the leakage current of the devices at low current density.*

[Response 6] We thank the reviewer for the comment on leakage current and efficiency of green phosphorescent OLEDs. As the reviewer noted, leakage current is critical to the overall device performance, and thus it is essential that devices exhibit low leakage current. To evaluate the leakage current of green phosphorescent OLEDs, linear plots of J - V - L curves were converted to logarithmic plots, as shown in Supplementary Fig. 13b. Low current density of under 10^{-6} A cm⁻² scale was observed for the non-operating voltage regime for all devices, with the typical diode characteristics of

J-V curves. Therefore, it is concluded that there was no leakage current of the devices at a low current density range. Nevertheless, the overall device performance of the green phosphorescent OLED was comparatively lower than that reported in other studies based on organic HIL and HTL used OLEDs. This can be understood by the increase of the exciton quenching probability due to the absence of the HTL, similar to the efficiency roll-off. To address these points, the blue phosphorescent OLED structure discussed earlier was designed to obtain high performance and large capability of the NiO:MoO₃ complex (Fig. 5). There were no leakage current, and 10⁻⁶ A cm⁻² scale and lower current densities were revealed from the logarithmic *J-V-L* plot of Supplementary Fig. 13e. The overall performance of the devices was significantly enhanced due to the proper design of the blue OLEDs, achieving a current efficiency of 32.6 cd A⁻¹ (~17% of EQE) in the device using NiO:MoO₃ 10 at.% HIL at 500 cd m⁻². This device efficiency is higher than that of the conventional HATCN HIL device (27 cd A⁻¹ at 500 cd m⁻²) and superior enough when comparing with other all-vacuum-deposited blue phosphorescent OLEDs (Fig 5 and Supplementary Fig. 12).¹ These consistent *J-V-L* tendencies across different optoelectronic configurations indicate that the NiO:MoO₃ complex can improve the charge balance by tailoring the energy band. In response to the reviewer's comment, we updated Fig. 4 and Fig. 5.

Reference

1. Lee J-H, *et al.* Blue organic light-emitting diodes: current status, challenges, and future outlook. *Journal of Materials Chemistry C* **7**, 5874-5888 (2019).

[Modifications in the manuscript]

(Results, page 12) “The detailed plots of each green and blue OLEDs were presented in Supplementary Fig. 13.”

(Supplementary Information)

Supplementary Fig. 12. Comparison of peak current efficiency and EQE of blue phosphorescent OLEDs corresponding to 10, 20, 30 at.% of NiO:MoO₃, and HATCN HIL. Mean value of current efficiency was 31.1, 29.0, 27.5, and 30.5 Cd A⁻¹ for NiO:MoO₃ 10 at.%, 20 at.%, 30 at.%, and HATCN, respectively. Average value of EQE was 16.1, 14.6, 13.7, and 15.6% for NiO:MoO₃ 10 at.%, 20 at.%, 30 at.%, and HATCN, respectively. Error bars were acquired from five OLED devices for each HIL type.

Supplementary Fig. 13. Detailed device characteristics of green and green and blue phosphorescent OLEDs. (a) Band structure of green phosphorescent OLED device and (b) logarithmic plot of J - V - L

(relative to Fig. 4). (a) Band structure of blue phosphorescent OLED device and (b) logarithmic plot of J - V - L (relative to Fig. 5).

[Comment 7] *A good HIL should have low surface roughness, and thus I suggest the authors to investigate the surface morphology especially the surface roughness of the NiO:MoO₃.*

[Response 7] We agree with the reviewer's comment that a uniform film surface is very important to ensure excellent interface formation for charge transport and injection. We examined the surface roughness by atomic force microscopy (AFM, XE-70 Park Systems) in non-contact mode with a super-sharp, tapping mode AFM probe (SSS-NCHR cantilever); NiO, NiO:MoO₃ complex, and MoO₃ films revealed extremely low surface roughness with root mean square (RMS) roughness values under 0.5 nm. Statements regarding surface roughness and the AFM data with RMS roughness values has been added to the revised manuscript and Supplementary Information.

[Modifications in the manuscript]

(Results, page 7) “Surface roughness investigated by atomic force microscopy (AFM) of each pristine and complex films exhibit excellent smoothness and uniformity. The root mean square (RMS) roughness values of all the samples were less than 0.5 nm, for example, an RMS roughness value of 0.17 nm for NiO:MoO₃ 50 at. % (Supplementary Fig. 3).”

(Methods, page 16) “Atomic force microscopy (AFM) was implemented with a XE-70 Park Systems in non-contact mode with SSS-NCHR cantilever.”

(Supplementary Information)

Supplementary Fig. 3. Surface roughness properties of each films. Atomic force microscopy (AFM) images ($1.5 \times 1.5 \mu\text{m}$ of scan area) of (a) NiO, (b) NiO:MoO₃ 15 at.%, (c) NiO:MoO₃ 30 at.%, (d) NiO:MoO₃ 50 at.%, (e) NiO:MoO₃ 70 at.%, (f) NiO:MoO₃ 85 at.%, and (g) MoO₃. (h) Their root mean square (RMS) roughness values are 0.286 nm, 0.198 nm, 0.191 nm, 0.168 nm, 0.173 nm, 0.185 nm, and 0.219 nm for NiO, NiO:MoO₃ 15 at.%, NiO:MoO₃ 30 at.%, NiO:MoO₃ 50 at.%, NiO:MoO₃ 70at.%, NiO:MoO₃ 85 at.%, and MoO₃, respectively.

[Comment 8] Fig. 4 (b), why NiO:MoO₃ with higher work function and higher conductivity shows less hole injection current?

[Response 8] This question regarding the device current density of NiO:MoO₃ complex HILs is related to the points discussed in [Response 4]. From the OLED structure in Fig. 4, the hole injection current is mainly determined by energy barriers at interfaces, which is directly affected by the increasing work function of NiO:MoO₃ complex HILs. Therefore, the barrier heights of both ITO/HIL and HIL/EML host interface should be considered to interpret the J - V - L characteristics, which are directly related with their energy band structure. For the 0 – 30 at.% ratio of MoO₃ NPs, the energy band of the NiO:MoO₃ complex drastically shifted to a deeper level, leading to a larger barrier height for hole injection; the barrier height

and work function were increased by 0.15 and 0.41 eV, respectively (Table R1). The reduced *J-V-L* characteristic of NiO:MoO₃ complex based OLEDs derived from these energy structure changes. A more important point to note is that the performance of the NiO:MoO₃ complex device was enhanced by improving the charge balance regardless of the hole injection current. Therefore, in the same manner as outlined in [Response 4], this *J-V-L* trend supports that a well-matched energy band structure is crucial for high performance of optoelectronics.

Table R1. Barrier height at interface of anode/HIL and HIL/EML host in green OLED configuration.

	NiO	NiO:MoO ₃ 15 at.%	NiO:MoO ₃ 30 at.%	MoO ₃
Work function (eV)	4.47	4.72	4.88	6.34
NiO E _v – ITO (eV)	0.67	0.72	0.82	-
MoO ₃ E _c – ITO (eV)	-	-0.47	-0.24	1.25
DCzppy – NiO E _v (eV)	0.68	0.63	0.53	
DCzppy – MoO ₃ E _c (eV)	-	1.59	0.99	0.1

Reviewer #2

[General remarks] *This paper reported an approach to improve the device performance of multilayer OLEDs by using a charge transfer complex (NiO:MoO₃-complex) as the hole injection layer. Although systematical characterizations on the structures and the charge transfer processes of NiO:MoO₃-complexes have been conducted in this research, the statement for performance improvement of the OLEDs based on NiO:MoO₃-complexes is unconvincing. Also, the authors didn't demonstrate the generality of this approach for improving the performance of OLEDs. Therefore, I cannot recommend the publication of this manuscript in Nat. Commun.*

[General response] We are grateful to the reviewer for the valuable comments that greatly helped us to improve our manuscript. In order to further clarify the concept of the NiO:MoO₃ complex system, detailed characterizations of the NiO:MoO₃ complex thin film and green phosphorescent OLEDs and were conducted. Furthermore, we demonstrated additional experiments based on blue phosphorescent OLEDs based on a Firpic EML to present both generality and improved OLED performance. In this regard, HATCN and other metal oxide-based approaches were also compared to the NiO:MoO₃ complex so that the advantages of our NiO:MoO₃ complex could be explained more clearly. We hope the revised version of our manuscript successfully delivers the concept of our work to the reviewer and readers of the journal.

[Comment 1] *In Figure 3f, it is noticed that the pure MoO₃ layer exhibited the highest conductivity among these complexes. How could the authors exclude the possibility that the improvement of the complex conductivity with the increased content of MoO₃ is due to the high conductivity of MoO₃?*

[Response 1] We would like to acknowledge the reviewer's insightful comment on electrical conductivity of the NiO:MoO₃ complex film. As the reviewer indicated, it is challenging to exclude the difference of electrical conductivity of NiO and MoO₃, which originates from physical properties of the materials.

Actually, the main point of Fig. 4f that we intended to emphasize is the particular trend in electrical conductivity with an increasing MoO₃ NPs ratio. The current density jumped by 79% for 15 at.% of MoO₃ NPs, and then shows a trend of saturated current density for the pristine MoO₃ film. Comparing our result with a previous report dealing with a mixture of two different materials (non-alloy) while excluding the interaction between materials or morphological effects, the electrical conductivities exhibited contrary trends; The electrical conductivity of the system was noticeably improved only when the proportion of higher conductivity materials was 50% or more.¹ We believe that this unusual trend is generated from our metal oxide heterostructure, which effectively induces charge transfer (Figs. 3a and 3b) and vacancy formation (Supplementary Fig. 6), thereby affecting the electrical conductivity of the metal oxide. In addition to the *J-V* results, we measured the electrical conductivity (resistivity), Hall mobility, and carrier concentration by a Hall effect measurement system to gain a more detailed understanding of the enhancement of conductivity in the NiO:MoO₃ complex. As shown in Supplementary Fig. 7, the electrical conductivity was enhanced sharply from $1.86 \cdot 10^{-4}$ S·cm of NiO to $2.93 \cdot 10^{-4}$ S·cm at NiO:MoO₃ 15 at.%, and increased to $4.24 \cdot 10^{-4}$ for MoO₃. The carrier concentration increased to a greater extent than the mobility decrement, contributing to the increasing trend of electrical conductivity in the NiO:MoO₃ complex thin film. This is noticeable because the results reinforce our argument that charge carriers were effectively generated by charge transfer from the NiO and MoO₃ complex structure, as well as by Ni vacancy creation during the fabrication process. In order to prevent any confusion of the readers, we added additional sentences that clarify that the unusual tendency of electrical conductivity in Fig. 3f is mainly induced by the formation of the charge transfer complex, even considering the electrical conductivity gap of the two materials.

Reference

1. Herissi L, Hadjeris L, Aida MS, Bougdira J. Properties of (NiO)_{1-x}(ZnO)_x thin films deposited by spray pyrolysis. *Thin Solid Films* **605**, 116-120 (2016).

[Modifications in the manuscript]

(Results, page 10) “The Hall effect measurement also provides a consistent trend of increasing conductivity for higher Mo fraction resulting from the enhanced carrier concentration (Supplementary Fig. 7). The increase in the conductivity is presumed to derive from further creation of charge carriers as a

consequence of the charge transfer and Ni^{2+} vacancies formation during the fabrication process. Vacancy generation ~”

(Methods, page 16) “Electrical resistivity (conductivity), Hall mobility, and carrier concentration were measured by Hall effect measurement system with the van der Pauw method (HMS Model 8407 including high resistance configuration, Lake Shore Cryotronics Inc.)”

(Supplementary Information)

Supplementary Fig. 7. Electrical conductivity trends of NiO:MoO₃ complex thin film. (a) Log J-log V plot of J - V characteristic (Fig. 3f) in ITO/thin film/Au device of NiO, MoO₃, and NiO:MoO₃ complex. Dashed lines are the linear fit of each device, following the ohmic conduction regime (slope = 1.00). (b) Comparison of resistivity conducted by the Hall measurement system and current density at 1 V from J - V curve (Fig. 3f) of NiO, MoO₃ and NiO:MoO₃ complex thin film. (c) Electrical conductivity, Hall mobility, and carrier concentration of NiO, MoO₃, and NiO:MoO₃ complex thin film obtained from Hall effect measurement.

[Comment 2] *The authors claimed that “the NiO:MoO₃ 30 at. %-complex HIL showed a substantially improved current efficiency (43% and 189%, respectively), due to well-defined energy structure formation.” However, the substantial improvement comes from self-comparison instead of the comparison with results reported in other works. Therefore, it is not so convincing. Especially, the performance of OLEDs with pristine NiO or MoO₃ HIL is very low and no statistic data is presented here.*

[Response 2] As the reviewer pointed out, the device performance demonstrated in Fig. 4 is ~5% of EQE, which is lower than those of typical all vacuum-deposited green phosphorescent OLEDs (~15%). Basically, we designed the device structure without HTL to identify the energy level modulation effect of the NiO:MoO₃ complex more definitively. The device structure with the NiO:MoO₃ complex showed a dramatic improvement of current efficiency; however, in turn, the probability of exciton quenching is promoted due to direct interface formation between the inorganic HIL and organic EML. In particular, NiO and MoO₃ based devices suffered more from poor charge balance in addition to exciton quenching, resulting in much lower current efficiency of 11.4 and 5.6 cd A⁻¹. Therefore, we fabricated and tested additional device structures to support our data and to resolve the reviewer’s concerns.

A blue phosphorescent OLED was designed with a structure of ITO/HIL/TAPC/TCTA-DCzppy:Firpic/BmPyPB/LiF/Al. TAPC HTL with high triplet energy, and TCTA-DCzppy:Firpic double EML were introduced to induce more efficient exciton formation and suppress the exciton quenching, leading to high device performance. The device structure, *J-V-L* characteristic, device efficiency, and spectra of the blue phosphorescent OLEDs are described in detail in Fig. 5. A comparison with the conventional HATCN HIL was also provided to verify the high capability and generality of the NiO:MoO₃ complex. Especially, NiO:MoO₃ 10 at.% complex based device shows high current efficiency of 32.6 cd A⁻¹ (at 500 cd m⁻²), which corresponds to a ~17% EQE. This device efficiency is higher than that of the conventional HATCN HIL device (27 cd A⁻¹ at 500 cd m⁻²) and superior enough when comparing with other all-vacuum-deposited blue phosphorescent OLEDs.¹ Statistic data of device performance in blue phosphorescent OLEDs are provided in Supplementary Fig. 12. Moreover, efficiency roll-off was significantly improved for all NiO:MoO₃ complex HILs (Supplementary Fig. 13). These consistent *J-V-L* tendencies across different optoelectronic configurations indicate the critical role of the NiO:MoO₃ complex in improving the charge balance by tailoring the energy band. The manuscript and Supplementary Information were updated to reflect the reviewer’s comment.

References

1. Lee J-H, *et al.* Blue organic light-emitting diodes: current status, challenges, and future outlook. *Journal of Materials Chemistry C* **7**, 5874-5888 (2019).

[Modifications in the manuscript]

(Results, page 12-13) “High capability and generality of NiO:MoO₃ complex. Finally, the NiO:MoO₃ complex was applied to another optoelectronic configuration to investigate its generality as a HIL. Blue phosphorescent OLEDs were demonstrated, with a structure of ITO/HIL/TAPC/TCTA-DCzppy:Firpic/BmPyPB/LiF/Al (Fig. 5a). A TAPC HTL with high triplet energy and a TCTA-DCzppy:Firpic double EML were introduced with the purpose of inducing more efficient exciton formation and high device performance. We also evaluated HATCN besides NiO:MoO₃ complex as a HIL, which is widely used in various OLED structures, to clarify the generality of the NiO:MoO₃ complex. The NiO:MoO₃ complex based devices show a similar *J-V-L* trend to those of the green phosphorescent OLED results above, revealing decreasing in current density and luminance with an increasing ratio of MoO₃ NPs (Fig. 5b). However, the overall device performance and efficiency roll-off characteristics were notably improved when compared to green phosphorescent OLEDs. In terms of device performance, high current efficiency of 32.6 cd A⁻¹ at 500 cd m⁻² (~17% of EQE) was achieved in the NiO:MoO₃ 10 at.% device which is even higher than that of the HATCN device (27 cd A⁻¹ at 500 cd m⁻², ~14% of EQE) (Fig. 5c and Supplementary Fig. 12). The detailed plots of each green and blue OLEDs were presented in Supplementary Fig. 13. As shown in Fig. 5d, consistent spectra of blue electroluminescence with 472 nm peak were observed for all devices. There were only slight spectrum shift in a broad range of 470 – 580 nm. In addition, it is found that the device prepared with the charge transfer complex HIL was advantageous over the other approaches discussed earlier, from comparisons using a blue phosphorescent OLED system (Supplementary Fig. 14).

Consistent tendencies in *J-V-L* characteristics across the different optoelectronic configurations indicate a definite operation of the NiO:MoO₃ complex with effective control of the electron-hole charge balance. Furthermore, we found that optimal device performances was obtained at a different MoO₃ fraction for each device structure. This implies that optimum device structures do not always result from a single

specific ratio of NiO:MoO₃ complex. Hence, the NiO:MoO₃ complex has huge potential for diverse optoelectronics such as a QLEDs (Supplementary Fig. 15) or PeLEDs, based on its general applicability due to the facile energy band tailoring ability.”

Fig. 5. Characterization of blue phosphorescent OLEDs. (a) Energy level diagram of blue phosphorescent OLED device. HATCN was performed as a comparison HIL with NiO:MoO₃ complex. (b) Current density-voltage-luminance curves, (c) luminance-current efficiency curves, (d) normalized electroluminescence for NiO:MoO₃ complex and HATCN based OLEDs. The inset image in (d) shows an operating OLED device.

(Methods, page 17) “Blue phosphorescent OLEDs were fabricated with a multilayer structure of ITO/HIL/TAPC(20 nm)/TCTA:Firpic (10 nm:7%)/DCzppy:Firpic (10 nm:10%)/BmPyPB (55 nm)/LiF(1 nm)/Al(100 nm). Processes except HIL deposition were sequentially thermally evaporated in ultrahigh vacuum ($P = 10^{-8}$ mbar), as done for green phosphorescent OLEDs.”

(Methods, page 17) “EQE was calculated by using the assumption of a Lambertian distribution of the emission.”

(Supplementary Information)

Supplementary Fig. 12. Peak current efficiency and EQE of blue phosphorescent OLEDs corresponding to 10, 20, 30 at.% of NiO:MoO₃, and HATCN HIL. Mean value of current efficiency was 31.1, 29.0, 27.5, and 30.5 Cd A⁻¹ for NiO:MoO₃ 10 at.%, 20 at.%, 30 at.%, and HATCN, respectively. Average value of EQE was 16.1, 14.6, 13.7, and 15.6% for NiO:MoO₃ 10 at.%, 20 at.%, 30 at.%, and HATCN, respectively. Error bars acquired from five OLED devices for each HIL types.

Supplementary Fig. 13. Detailed device characteristics of green and green and blue phosphorescent OLEDs. (a) Band structure of green phosphorescent OLED device and (b) logarithmic plot of J - V - L (relative to Fig. 4). (a) Band structure of blue phosphorescent OLED device and (b) logarithmic plot of J - V - L (relative to Fig. 5).

[Comment 3] *The authors claimed that they could induce effective energy level modulation over a wide work function range of 4.47-6.34 eV. How about the general applicability of using the NiO:MoO₃-complex to improve the performance of OLEDs based on other luminescent materials besides DCzppy-Ir(ppy)₃?*

[Response 3] We are grateful to receive the insightful comment about the generality of the NiO:MoO₃ complex. As described in our manuscript, one of the key advantages of the NiO:MoO₃ complex system is energy level modulation ability over a wide work function range. This property provides remarkable potential in terms of applicability to various optoelectronic systems. In this regard, another light-emitting material system of TCTA-DCzppy:Firpic was introduced as an EML of OLEDs. Firpic is a representative blue-emitting phosphorescent material and was incorporated into the bilayer structure of TCTA and

DCzppy to induce facile exciton formation. The designed device structure of the blue phosphorescent OLED was ITO/HIL/TAPC/TCTA-DCzppy:Firpic/BmPyPB/LiF/Al, which is the same as that of the device discussed in Response 2 (Fig. 5). We adopted this design to demonstrate the more general effectiveness of the NiO:MoO₃ complex via testing additional OLED devices based on other luminescent materials. From the results in Fig. 5, we noticed that not only does the NiO:MoO₃ complex HIL operate as well as the conventional HATCN HIL, but also the device efficiency was maximized with the NiO:MoO₃ complex HIL of 10 at.% of MoO₃ NPs. Considering that the NiO:MoO₃ 30 at.% HIL achieved the highest efficiency in green phosphorescent OLEDs, it could be concluded that providing a tunable energy level of the HIL for a given energy structure is highly advantageous because the optimum composition of NiO:MoO₃ complex varies for different device structures, suggesting its general applicability based on large controllability of energy levels.

In addition, we performed a comparative experiment based on the device structure of blue phosphorescent OLEDs. As shown in Supplementary Fig. 14, three types of devices were fabricated, where different HILs with a charge transfer complex (NiO:MoO₃ 10 at.%), conventional doping (NiO:Mo 10 at.%), and SCTD methods (NiO/MoO₃ bilayer) were employed. Conventional doping was performed by the same methods as employed for NiO film fabrication but Mo acetate dimer was adopted instead of Ni acetate tetrahydrate as a precursor. The SCTD system was made by bilayer formation of NiO and MoO₃ thin films. The device with the charge transfer complex HIL shows relatively low *J-V-L* characteristic when compared to the conventional doping HIL based device, and the SCTD HIL device follows with the lowest *J-V-L* characteristics. For the device performance, the charge transfer complex HIL exhibits the highest current efficiency (27.7 cd A⁻¹ at 1000 cd m⁻²) along with suppressed efficiency roll-off while the conventional doping HIL shows lower current efficiency and a poor efficiency roll-off property. This is because the NiO:MoO₃ complex provides more favorable charge balance in the OLED device. The device based on the SCTD HIL suffered from a disadvantageous band structure, resulting in the lowest current efficiency. In summary, we confirmed the comparative advantage of our charge transfer complex system by comparing it with other approaches, and also showed its high capability based on excellent performance in blue phosphorescent OLEDs.

Furthermore, application of the NiO:MoO₃ complex to quantum dot light-emitting diodes (QLEDs) was attempted to show the expandability of the NiO:MoO₃ complex. The device structure was ITO/NiO:MoO₃ complex/PVK/CdSe QDs/ZnO/Al, which is one of the typical normal structure of QLEDs. As shown in Supplementary Fig. 15, a moderate level of *J-V-L* characteristic along with a peak current efficiency of

2.95 cd A⁻¹ at 1000 cd m⁻² was revealed. Also, a narrow green emission of 525 nm supports the fine operation of the NiO:MoO₃ complex as a HIL in QLEDs. These points indicate that improved *J-V-L* characteristic and performance are attainable by additional optimization. For example, we expect that better device performance can be achieved by gradient energy band formation via stacking charge transfer complex thin films. Additional data and sentences were provided in the revised manuscript and Supplementary Information to address the reviewer’s comment.

[Modifications in the manuscript]

(Results, page 12) “In addition, it is found that the device prepared with the charge transfer complex HIL was more advantageous over the other approaches discussed earlier (also see comparisons of blue phosphorescent OLED system in Supplementary Fig. 14).”

(Results, page 13) “Hence, the NiO:MoO₃ complex have huge potential to diverse optoelectronics such as a QLEDs (Supplementary Fig. 15) or PeLEDs, based on its general applicability due to the facile energy band tailoring ability.”

(Supplementary Information)

Supplementary Fig. 14. Comparison of metal oxide enhancing strategies in blue phosphorescent

OLED system. Schematics of energy band and charge balance of (a) charge transfer complex (this work), (b) conventional doping, and (c) surface charge transfer doping (SCTD). (d) Current density-voltage-luminance curves, (e) luminance-current efficiency curves, and (f) normalized electroluminescence for devices prepared by charge transfer complex (NiO:MoO₃ 10 at.%), conventional doping (NiO:Mo 10 at.%), and SCTD (NiO/MoO₃ bilayer).

Supplementary Fig. 15. Characterization of quantum dot light emitting diodes (QLEDs). (a) Energy level diagram of multilayer QLED device. (b) Current density-voltage and luminance curves as a function of applied voltage, (c) luminance-current efficiency curves, (d) normalized electroluminescence for NiO:MoO₃ complex QLEDs. The dashed line in (b) is the polynomial fit curve of luminance. The inset image in (d) shows an operating QLED device.

Methods

QLED device fabrication.

The ITO-patterned glass substrates (resistivity: $< 20 \text{ ohm square}^{-1}$) were cleaned by sonication with acetone, ethanol, and isopropyl alcohol for 30 min each, followed by UV-O₃ treatment for 20 min. The QLED devices were fabricated by spin-coating NiO:MoO₃ complex, PVK (TCI), QDs (Uniam), and ZnO in order, except for the Al cathode. After spin-casting of NiO:MoO₃ 10 at.% complex (3000 rpm, 30 s), the sample was annealed at 300 °C for 1 hour under ambient conditions and was then moved to a N₂ gas filled glove box to form the next layers. PVK (8 mg mL⁻¹ in chlorobenzene) was spin-coated at 3000 rpm for 60 s and baked at 180 °C for 30 min, followed by QD solution (15 mg mL⁻¹ in octane) coating at 3000 rpm for 30 seconds and baking for 30 min at 80 °C. ZnO NPs (30 mg mL⁻¹ in 2-methoxyethanol) were deposited by spin-coating (3000 rpm for 60 seconds) onto the QD layer. Finally, a 120 nm Al cathode was thermally evaporated under high vacuum conditions (10⁻⁸ mbar).

Synthesis of ZnO NPs.

To synthesize ZnO NPs, 1.8 mmol of zinc acetate dihydrate (Zn(OAc)₂ · 2H₂O) was loaded in a 100 mL flask with 20 ml of ethanol. The Zn precursor solution was stirred for 1 h at 80 °C. After adding 2.8 mmol of lithium hydroxide monohydrate (LiOH · H₂O), the flask was sonicated for 1 h to synthesize ZnO NPs. Finally, the synthesized ZnO NP solution was purified by centrifugation with hexane mixed into the solution.

Reviewer #3

[General remarks] *From my point of view, this manuscript show results which are worthy of publication in Nature Communications. However, some corrections had to be made previous to the acceptance of the paper for publication. These changes are related with the next comments:*

[General response] We appreciate the reviewer's positive evaluation of the manuscript. Additional information and corrections were made to address the reviewer's comments. Especially, a comparison of the charge transfer complex system with conventional doping and surface charge transfer doping systems, the size distribution of MoO₃ NPs, and further study in the blending process of NiO and MoO₃ NPs solution were accomplished. Furthermore, we conducted additional experiments based on blue phosphorescent OLEDs based on Firpic EML to present both generality and improved OLED performance. In this regard, HATCN and other metal oxide enhancing approaches were also compared to the NiO:MoO₃ complex so that the advantages of the NiO:MoO₃ complex could be seen more clearly. We hope the revised version of our manuscript successfully delivers the concept of our work to the reviewer and readers of the journal.

[Comment 1] *I think that "complex" is not the correct word that should be use to define these materials. They are composite materials formed by nanoparticles immersed in a matrix. Therefore, I think "complex" should be changed by "composite. Actually, the material is not a NiO:MoO₃ NPs "heterostructure" (not different layers in the film), but a composite thin film material of a NiO matrix with MoO₃ NPs.*

[Response 1] We thank the reviewer for the detailed comment regarding the terminology of “complex” and “heterostructure”. The phrasing of metal oxide charge transfer complex is inspired by organic charge transfer complex^{1,2} which is a blended form of two organic molecules (electron donor and acceptor). In their system, it is called a “complex” despite that two molecules are not actually bonded and they exist as a mixture form. The charge transfer complex exhibits extraordinary behavior by interaction (charge

transfer) of the components within the scale of a few nanometers. In the dictionary sense, composite and complex are distinguished by whether there is a connection between substances (composite: something that is made of various different parts; complex: made of a lot of different but connected parts; from Cambridge Dictionary). From our interpretation and known content, the suggested concept seems more appropriate to be called a “complex” because two metal oxides have a strong connection within a few nanometer scale and produce the charge transfer phenomenon, which is one of the most important points of our concept. The expression “heterostructure” is also used in the same context as “complex”. In our manuscript, the interface formation of two metal oxides within a nanometer scale is crucial to induce the charge transfer phenomenon and this structure is demonstrated by a particular methodology and process conditions. Also, from several reports, we noticed that “heterostructure” is not only accessible to certain systems with different layers in the film, but also available to systems having an interface of dissimilar materials with any combinations of forms.^{3,4,5} On the basis of these reasons, we perceive “heterostructure” as a suitable expression for making our manuscript more convincing to readers of the journal.

References

1. Ferraris J, Cowan DO, Walatka V, Perlstein JH. Electron transfer in a new highly conducting donor-acceptor complex. *Journal of the American Chemical Society* **95**, 948-949 (1973).
2. Goetz KP, Vermeulen D, Payne ME, Kloc C, McNeil LE, Jurchescu OD. Charge-transfer complexes: new perspectives on an old class of compounds. *Journal of Materials Chemistry C* **2**, 3065-3076 (2014).
3. Chen P-C, *et al.* Interface and heterostructure design in polyelemental nanoparticles. *Science* **363**, 959-964 (2019).
4. Lauhon LJ, Gudiksen MS, Wang D, Lieber CM. Epitaxial core-shell and core-multishell nanowire heterostructures. *Nature* **420**, 57-61 (2002).
5. Jariwala D, Marks TJ, Hersam MC. Mixed-dimensional van der Waals heterostructures. *Nature Materials* **16**, 170-181 (2017).

[Comment 2] Pg.10. *The efficiencies obtained in these materials should be compared with those of materials of these metal oxides prepared by the doping method and with the SCTD. It would be good to show a table or graphic showing results obtained using other preparation concepts compared with the*

results obtained in this work for different NiO/MoO₃ NPs ratios.

[Response 2] We appreciate the reviewer's detailed comment for a comparison with other metal oxide enhancing approaches. Clarifying the advantages of the NiO:MoO₃ complex is crucial, and therefore we agree that a comparison with the conventional doping and SCTD methods is appropriate. Before conducting the suggested experiment, we designed an additional optoelectronic structure based on a blue phosphorescent material to render high compatibility and generality of the NiO:MoO₃ complex (Fig. 5). A blue phosphorescent OLED was designed with a structure of ITO/HIL/TAPC/TCTA-DCzppy:Firpic/BmPyPB/LiF/Al. A TAPC HTL with high triplet energy and a TCTA-DCzppy:Firpic double EML were introduced to induce more efficient exciton formation and suppress the exciton quenching, leading to high device performance with improved efficiency roll-off characteristics. Furthermore, characterization of blue phosphorescent OLEDs was provided including a comparison with the conventional HATCN HIL. Especially, NiO:MoO₃ 10 at.% complex based device shows high current efficiency of 32.6 cd A⁻¹ (at 500 cd m⁻²), which corresponds to a ~17% EQE. This device efficiency is higher than that of the conventional HATCN HIL device (27 cd A⁻¹ at 500 cd m⁻²) and superior enough when comparing with other all-vacuum-deposited blue phosphorescent OLEDs.¹ Statistic data of device performance in blue phosphorescent OLEDs are provided in Supplementary Fig. 12. Moreover, efficiency roll-off was significantly improved for all NiO:MoO₃ complex HILs (Supplementary Fig. 13). These consistent *J-V-L* tendencies across different optoelectronic configurations indicate the critical role of the NiO:MoO₃ complex in improving the charge balance by tailoring the energy band.

We then performed the suggested comparison experiment based on the device structure of the blue phosphorescent OLED. As shown in Supplementary Fig. 14, three types of devices were fabricated, which have varied HILs with charge transfer complex (NiO:MoO₃ 10 at.%), conventional doping (NiO:Mo 10 at.%) and SCTD methods (NiO/MoO₃ bilayer). Conventional doping was carried out by the same methods as used in NiO film fabrication but with the adoption of Mo acetate dimer instead of Ni acetate tetrahydrate as a precursor. The SCTD system was made by bilayer formation of NiO and MoO₃ thin films. The device with the charge transfer complex HIL shows a relatively low *J-V-L* characteristic when compared to the conventional doping HIL based device, and the SCTD HIL device follows with the lowest *J-V-L* characteristics. For the device performance, the charge transfer complex HIL exhibits the highest current efficiency (27.7 cd A⁻¹ at 1000 cd m⁻²) along with suppressed efficiency roll-off while the conventional doping HIL shows lower current efficiency and a poor efficiency roll-off property. This is because the NiO:MoO₃ complex provides more favorable charge balance in the OLED device. The device

based on the SCTD HIL suffered from a disadvantageous band structure, resulting in the lowest current efficiency. In summary, we confirmed the comparative advantage of our charge transfer complex system by comparing it with other approaches, while also revealing high capability based on excellent performance in blue phosphorescent OLEDs. Additional data and sentences were provided in the manuscript and Supplementary Information to address the reviewer's comment.

Reference

1. Lee J-H, *et al.* Blue organic light-emitting diodes: current status, challenges, and future outlook. *Journal of Materials Chemistry C* **7**, 5874-5888 (2019).

[Modifications in the manuscript]

(Results, page 12) “In terms of device performance, high current efficiency of 32.6 cd A^{-1} at 500 cd m^{-2} (~17% of EQE) was achieved in NiO:MoO₃ 10 at.% device which is even higher than that of HATCN device (27 cd A^{-1} at 500 cd m^{-2} , ~14% of EQE) (Fig. 5c and Supplementary Fig. 12).”

(Results, page 12) “The detailed plots of each green and blue OLEDs were presented in Supplementary Fig. 13.”

(Results, page 12) “In addition, it is found that the device prepared with the charge transfer complex HIL was more advantageous over the other approaches discussed earlier (also see comparisons of blue phosphorescent OLED system in Supplementary Fig. 14).”

Fig. 5. Characterization of blue phosphorescent OLEDs. (a) Energy level diagram of blue phosphorescent OLED device. HATCN was performed as a comparison HIL with NiO:MoO₃ complex. (b) Current density-voltage-luminance curves, (c) luminance-current efficiency curves, (d) normalized electroluminescence for NiO:MoO₃ complex and HATCN based OLEDs. The inset image in (d) shows an operating OLED device.

(Supplementary Information)

Supplementary Fig. 12. Comparison of peak current efficiency and EQE of blue phosphorescent

OLEDs corresponding to 10, 20, 30 at.% of NiO:MoO₃, and HATCN HIL. Mean value of current efficiency was 31.1, 29.0, 27.5, and 30.5 Cd A⁻¹ for NiO:MoO₃ 10 at.%, 20 at.%, 30 at.%, and HATCN, respectively. Average value of EQE was 16.1, 14.6, 13.7, and 15.6% for NiO:MoO₃ 10 at.%, 20 at.%, 30 at.%, and HATCN, respectively. Error bars acquired from five OLED devices for each HIL types.

Supplementary Fig. 13. Detailed device characteristics of green and blue phosphorescent OLEDs. (a) Band structure of green phosphorescent OLED device and (b) logarithmic plot of J - V - L (relative to Fig. 4). (d) Band structure of blue phosphorescent OLED device and (e) logarithmic plot of J - V - L (relative to Fig. 5).

Supplementary Fig. 14. Comparison of metal oxide enhancing strategies in blue phosphorescent OLED system. Schematics of energy band and charge balance of (a) charge transfer complex (this work), (b) conventional doping, and (c) surface charge transfer doping (SCTD). (d) Current density-voltage-luminance curves, (e) luminance-current efficiency curves, and (f) normalized electroluminescence for devices prepared by charge transfer complex (NiO:MoO₃ 10 at.%), conventional doping (NiO:Mo 10 at.%), and SCTD (NiO/MoO₃ bilayer).

[Comment 3] Fig.4. Differences in the results of electroluminicence obtained for NiO, NiO_MoO₃-complex and MoO₃ are not clearly observed Fig 4d. Modify this figure to see at a first sight the differences among the materials.

[Response 3] We are grateful for the reviewer for suggesting the modification of Fig. 4d to improve its visibility. The spectral shape of electroluminescence (EL) is a key property that determines color coordination, which represents the visualized color itself. Providing consistency in terms of the EL spectrum is important to produce the initially intended color, but the EL spectrum could be altered due to many reasons such as formation of an exciplex, micro-cavity effect, and recombination zone shift.^{1,2,3}

Therefore, many studies that control electron-hole charge balance often display consistency of the EL spectra as evidence that their concept or strategy only affects charge injection or transport properties. For the same purpose, we presented the EL spectra of devices using different fractions of MoO₃ NPs to address the consistency of the EL spectrum in Fig. 4d. The exciplex peak at around 450 nm was not observed, and only a negligible shift was noted from magnified EL spectra (Fig. R2). However, a slight shift in EL spectra at 470 – 580 nm was noticed in the blue phosphorescent OLEDs (Fig. 5d). We infer that the change in the EL spectrum is caused by a shift of the recombination zone, considering that the EL shift occurs in a broad range of 470 – 580 nm. Therefore, an additional statement for Fig. 5d is provided for better understanding.

Fig. R2. Magnified electroluminescence spectra of NiO, MoO₃, and NiO:MoO₃ complex based green phosphorescent OLED devices.

References

1. Tang X, *et al.* Highly efficient luminescence from space-confined charge-transfer emitters. *Nature Materials* **19**, 1332-1338 (2020).
2. Jesuraj PJ, *et al.* Recombination Zone Control without Sensing Layer and the Exciton Confinement in Green Phosphorescent OLEDs by Excluding Interface Energy Transfer. *The Journal of Physical Chemistry C* **122**, 2951-2958 (2018).
3. Seo H-K, *et al.* Efficient Flexible Organic/Inorganic Hybrid Perovskite Light-Emitting Diodes Based on Graphene Anode. *Advanced Materials* **29**, 1605587 (2017).

[Modifications in the manuscript]

(Results, page 12) “There were only slight spectrum shift in a broad range of 470 – 580 nm.”

[Comment 4] *Discussion. Actually, this is the conclusion section. Here, discussion of the results is not made. It is made in the previous section of Results. Therefore, you can do two things if the format of the journal allow you to do it: 1st. Include two different sections: "Results" and "Discussion", separating and moving the discussion of the results now done in the "Results" section to the "Discussion" section. 2nd. Changing the caption of the Results section by "Results and Discussion". And then, the "Conclusions".*

[Response 4] We understand the reviewer’s concern about the formatting of section headings and checked the formatting instructions of Nature Communications to address this. Regretfully, section headings in the main text are only permitted to use “Results” and “Discussion”, not including “Conclusions”. We comprehend the “Discussion” section as describing the discussions from the perspective of the overall manuscript. It is also confirmed that the recent articles of Nature Communications consistently follow this guideline. Therefore, we have to maintain the section heading in the current manuscript form to adhere to the guidelines of the journal.

[Comment 5] *Methods. In "Synthesis of NiO precursor solution and MoO₃ nanoparticles", Results should be provided in this section about the characterization of the MoO₃ NPs. Mainly, average particle size and distribution of particle sizes. Crystallinity is correctly shown in Fig 2 by XRD, as a function of the annealing temperature.*

[Response 5] We appreciate the reviewer’s valuable comment. To verify the particle distribution of a few nanometer scale MoO₃, we conducted HAADF-STEM to obtain TEM images with much higher

resolution than typical HRTEM (Fig. R3 and Supplementary Fig. 2). The HAADF-STEM image is converted to a monochromic image by threshold adjustment of image analysis software (ImageJ), and a statistical particle analysis was performed. The synthesized MoO₃ NPs show a monodisperse size distribution with a 2.9 nm median, which is effective to induce charge transfer between NiO and MoO₃ because the charge transfer phenomenon occurs within ~10 nm of the interface (Supplementary Fig. 1). In response to the reviewer's comment, we provide additional data of MoO₃ NPs and their size distribution in the Methods section.

Fig. R3. Size and distribution of monodisperse MoO₃ NPs. HAADF-STEM image of (a) MoO₃ NPs (scale bar: 20 nm). (b) Monochromic MoO₃ NPs image (scale bar: 20 nm) converted by ImageJ software and (c) their size distribution histogram.

[Modifications in the manuscript]

(Methods, page 15) “MoO₃ nanoparticles have a monodisperse distribution with a 2.9 nm median (Supplementary Fig. 2).”

(Methods, page 16) “Structural characteristics of NiO, NiO:MoO₃ NPs complex, and MoO₃ NPs were investigated by transmission electron microscopy (HRTEM: JEM-2100F by JEOL, and HAADF-STEM: Talos F200X by FEI)”

(Supplementary Information)

Supplementary Fig. 2. Size and distribution of monodisperse MoO₃ NPs. HRTEM images of (a) MoO₃ NPs (scale bar: 20 nm) and (b) a magnified image (scale bar: 2 nm) showing the lattice fringes of the NPs. HAADF-STEM image of (c) MoO₃ NPs (scale bar: 20 nm) and their size distribution histogram.

[Comment 6] Methods. In "Preparation of NiO:MoO₃-complex films", How do you assure that agglomeration of the MoO₃ NPs is not produced? A simple blending of the NiO solution and the MoO₃ NPs is difficult that avoid nanoparticles agglomeration. Indicate this in this section. Also here, Indicate the thickness of the layer that is obtained after the deposition of the mixture solutions.

[Response 6] We thank the reviewer for the insightful comment regarding the possible agglomeration by blending two solutions. As pointed out in the comments, a simple blending of the NiO precursor and MoO₃ NPs solution suffer from aggregation of MoO₃ NPs. Therefore, we added H₂O₂ for 0.4 M in MoO₃ solution before conducting the complexing process so that H₂O₂ acts as a stabilizer to prevent any agglomeration of NPs or precursors (Supplementary Fig. 16). We presume that the remaining excess ethanolamine (MEA) after NiO precursor complexation¹ induces reaction with the -OH ligand of MoO₃ NPs, leading to immediate agglomeration of the Ni precursor and MoO₃ NPs complex solution. H₂O₂ is introduced to hinder agglomeration by providing extra -OH and -OOH ligands of MoO₃ NPs.² As shown in Supplementary Fig. 17, agglomeration occurs immediately when blending the MoO₃ NPs solution without H₂O₂ with the NiO precursor solution, and H₂O₂ effectively stabilizes the complex solution as the concentration increases. Thus, we use a 0.4 M H₂O₂ complex solution with good stability over a few days. Although we included the information about adding H₂O₂ at the first sentence of “Preparation of NiO:MoO₃-complex films” in Methods, but it is not enough to understand the role and details of H₂O₂. Supplementary sentences and figures are added in the Methods section for better understanding. Also, a Supplementary Movie 1 was created for more intuitive visualization. Regarding the film thickness immediately after deposition, the film after the spin-coating process not only includes the Ni precursor and MoO₃ NPs but also the solvent and moisture. For this reason, it is difficult to observe the thickness of the film after deposition including all components by a method such as SEM imaging. Ellipsometry was carried out to measure the film thickness using the Cauchy model. The thickness of the layer after spin-coating of the complex solutions was 91.4 – 86.9 nm, which is newly added to Methods section.

References

1. Manders JR, *et al.* Solution-Processed Nickel Oxide Hole Transport Layers in High Efficiency Polymer Photovoltaic Cells. *Advanced Functional Materials* **23**, 2993-3001 (2013).
2. Kadossov EB, Soufiani AR, Apblett AW, Materer NF. Density-functional studies of hydrogen peroxide adsorption and dissociation on MoO₃(100) and H_{0.33}MoO₃(100) surfaces. *RSC Advances* **5**, 97755-97763 (2015).

[Modifications in the manuscript]

(Methods, page 15) “H₂O₂ acts as a stabilizer from the agglomeration of MoO₃ NPs in a blending solution of Ni precursor and MoO₃ NPs (Supplementary Fig. 16, 17 and Supplementary movie 1). A

MoO₃ NP solution with H₂O₂ was then added to the NiO precursor solutions and blended with a Vortex mixer for 1 minute.”

(Methods, page 16) “The film thickness after spin-coating was 91.4 – 86.9 nm, and the thickness decreased to 22.4 – 20.3 nm by the annealing process. Ellipsometry characterization combined with the Cauchy model was introduced to measure the film thickness containing solvent and moisture.”

(Supplementary Information)

Supplementary Fig. 16. H₂O₂ as a stabilizer in NiO precursor and MoO₃ NP complexing process. The complex solution with 0.4 M of H₂O₂ as a stabilizer shows a clear green color while the simple blended solution was opaque and agglomerated. H₂O₂ is introduced to hinder agglomeration by providing extra -OH and -OOH ligands of MoO₃ NPs.

Supplementary Fig. 17. Stability of the complex solution depending on the added H₂O₂ concentration. Snapshots of Supplementary Movie 1 at certain times immediately after blending the two solutions are provided.

(Supplementary Files)

Supplementary Movie 1 has been added.

[Comment 7] *Methods. In "Fabrication and measurements of OLEDs". Spelling mistakes, it should be 300 °C. Also, here indicate the thickness of the layer after the annealing.*

[Response 7] We appreciate the reviewer's careful comment. The use of the special character Celsius degrees (°C) seems to have made a mistake, visualized by the broken symbol (□) that occurred during the pdf conversion process. We also fixed the spelling mistakes on Celsius degrees by adjusting the proper special character. With regard to the film thickness after the annealing process, the solution-processed films exhibit decreased thicknesses after the annealing process, which is a result of film condensation due to evaporation of the solvent.¹ Film thickness was evaluated by an ellipsometer combined with the Cauchy model. The film thickness of NiO, NiO:MoO₃ complex, and MoO₃ after the annealing process was in a range of 22.4 – 20.3 nm. Thickness information is provided in the Methods section of our manuscript.

References

1. Girotto C, Voroshazi E, Cheyns D, Heremans P, Rand BP. Solution-Processed MoO₃ Thin Films As a Hole-Injection Layer for Organic Solar Cells. *ACS Applied Materials & Interfaces* **3**, 3244-3247 (2011).

[Modifications in the manuscript]

(Methods, page 17) The special character display error of "°C" has been corrected.

(Methods, page 16) “The film thickness after spin-coating was 91.4 – 86.9 nm, and the thickness decreased to 22.4 – 20.3 nm by the annealing process. Ellipsometry characterization combined with the Cauchy model was introduced to measure the film thickness containing solvent and moisture.”

[Other modifications in the manuscript]

(Authors) The authorship and author order has been changed in consideration of contribution to the revised version of the manuscript.

- Initial version of manuscript: Moohyun Kim, Myeong Seon Cho, **Byoung-Hwa Kwon, Chul Woong Joo**, Ye ji Kim, Hyunjin Cho, **Hanhwi Jang**, Duk Young Jeon, Eugene N. Cho*, and Yeon Sik Jung*

- Current version of manuscript: Moohyun Kim, **Byoung-Hwa Kwon, Chul Woong Joo**, Myeong Seon Cho, **Hanhwi Jang**, Ye ji Kim, Hyunjin Cho, Duk Young Jeon, Eugene N. Cho*, and Yeon Sik Jung*
(These authors contributed equally: Moohyun Kim, Byoung-Hwa Kwon)

(Authors) Typo in author affiliation has been corrected from “..., Daehak-ro, ...” to “..., **291** Daehak-ro, ...”.

Reviewers' Comments:

Reviewer #1:

Remarks to the Author:

Although the authors have revised the paper and addressed some of my concerns, my main concern of the advantages of the proposed NiO:MoO₃ over conventional hole-injection materials remains unaddressed. Even with NiO:MoO₃, the device performance is relatively low compared to that of state-of-the-art devices, and the improvement in efficiency is limited. Also, there are no stability data showing whether the lifetime of the devices could be improved by using NiO:MoO₃. I agree with the comments of reviewer 2 that the performance improvement of the OLEDs based on NiO:MoO₃-complexes is unconvincing. Considering the limited novelty and the advantage of NiO:MoO₃, I cannot recommend the publication of this manuscript in Nature Communications.

Reviewer #2:

Remarks to the Author:

In the revised manuscript, the authors have successfully addressed most of the reviewers' questions. I feel the manuscript has been remarkably improved and could be acceptable in its current form.

Reviewer #3:

Remarks to the Author:

From my point of view the authors have addressed properly all of my comments. In addition, they have highly improve the quality of the paper by including new results that provide interesting information about the potential application of these multilayer metal oxide complex in optoelectronic devices.

Therefore, I consider that this work can be accepted for publication in Nature Communications without including additional changes.

■ Point-by-point responses to the reviewers' comments.

We provide point-by-point responses to the reviewers' in-depth comments, and the updated text in the revised manuscript is marked in blue.

Reviewer #1

[General remarks] *Although the authors have revised the paper and addressed some of my concerns, my main concern of the advantages of the proposed NiO:MoO₃ over conventional hole-injection materials remains unaddressed. Even with NiO:MoO₃, the device performance is relatively low compared to that of state-of-the-art devices, and the improvement in efficiency is limited. Also, there are no stability data showing whether the lifetime of the devices could be improved by using NiO:MoO₃. I agree with the comments of reviewer 2 that the performance improvement of the OLEDs based on NiO:MoO₃-complexes is unconvincing. Considering the limited novelty and the advantage of NiO:MoO₃, I cannot recommend the publication of this manuscript in Nature Communications.*

[General response] We appreciate the reviewer's detailed comments for the manuscript. In this study, we introduced the novel concept of the metal oxide charge transfer complex and their specific demonstration with NiO:MoO₃ complex. We performed systematic analyses to elucidate the underlying mechanism and to validate our hypothesis. We presented that nanodomain formation inside the matrix phase via thermodynamic-control of sol-gel process can substantially modulate both charge transport and injection capabilities without limiting doping concentration. The scope of this study is not restricted to fundamental characterization of the charge transfer complex because we also confirmed that the NiO:MoO₃ complex can significantly improve the OLED performance. As Reviewer #1 mentioned, Reviewer #2 requested additional data regarding the device performance, and thus we performed extensive experiments during the last several months and finally provided significantly improved data. We believe that Reviewer #2 acknowledged the updated data by commenting that *"In the revised manuscript, the authors have successfully addressed most of the reviewers' questions. I feel the manuscript has been remarkably improved and could be acceptable in its current form."* However, we additionally modified the manuscript to clarify the key outcomes of this study in terms of both material characteristics and device performances

as follows.

[Modifications in the manuscript]

(Abstract, page 2) “However, conventional methods cannot enable both energy level manipulation and conductivity enhancement for achieving optimum energy band configurations. Here, we introduce a metal oxide charge transfer complex (NiO:MoO₃-complex), which is composed of few-nm-size MoO₃ domains embedded in NiO matrices, as a highly tunable carrier injection material. Charge transfer at the finely dispersed interfaces of NiO and MoO₃ throughout the entire film enables effective energy level modulation over a wide work function range of 4.47 – 6.34 eV along with enhanced electrical conductivity. The high performance of NiO:MoO₃-complex is confirmed by achieving 189% improved current efficiency compared to that of MoO₃-based green OLEDs and also an external quantum efficiency of 17% when applied to blue OLEDs, which is superior to 1,4,5,8,9,11-hexaazatriphenylene-hexacarbonitrile-based conventional devices.”

(Introduction, page 4) “The NiO:MoO₃ NPs heterostructure charge transfer complex (NiO:MoO₃-complex) exhibits extensive controllability of the work function in a range of 4.47 – 6.34 eV and electrical conductivity improvement of up to 2.4 times compared to that of pristine NiO, without MoO₃ concentration limit. The NiO:MoO₃-complex realizing 43% and 189% increased current efficiency in the green phosphorescent OLED system relative to those of pristine NiO and MoO₃ by achieving a well-configured energy band structure for excellent electron-hole charge balance.”

Reviewer #2

[General remarks] *In the revised manuscript, the authors have successfully addressed most of the reviewers' questions. I feel the manuscript has been remarkably improved and could be acceptable in its current form.*

[General response] We appreciate the reviewer’s positive evaluation for the revised version of

manuscript. The reviewer's insightful comments and suggestions greatly helped us to improve our manuscript. We hope the final version of the manuscript successfully delivers the concept of our work to the readers of the journal.

Reviewer #3

[General remarks] *From my point of view the authors have addressed properly all of my comments. In addition, they have highly improve the quality of the paper by including new results that provide interesting information about the potential application of these multilayer metal oxide complex in optoelectronic devices.*

Therefore, I consider that this work can be accepted for publication in Nature Communications without including additional changes.

[General response] We appreciate the reviewer's positive review of the revised version of manuscript. The reviewer's valuable comments and suggestion enabled us to greatly improve our manuscript. We hope the final version of the manuscript successfully delivers the concept of metal oxide charge transfer complex to the readers of the journal.